# SPARSE QUANTIZED SPECTRAL CLUSTERING

**Zhenyu Liao**
ICSI and Department of Statistics
University of California, Berkeley, USA
zhenyu.liao@berkeley.edu

**Romain Couillet**
G-STATS Data Science Chair, GIPSA-lab
University Grenobles-Alpes, France
romain.couillet@gipsa-lab.grenoble-inp.fr

**Michael W. Mahoney**
ICSI and Department of Statistics
University of California, Berkeley, USA
mmahoney@stat.berkeley.edu

## ABSTRACT

Given a large data matrix, sparsifying, quantizing, and/or performing other entry-wise nonlinear operations can have numerous benefits, ranging from speeding up iterative algorithms for core numerical linear algebra problems to providing nonlinear filters to design state-of-the-art neural network models. Here, we exploit tools from random matrix theory to make precise statements about how the eigenspectrum of a matrix changes under such nonlinear transformations. In particular, we show that very little change occurs in the informative eigenstructure, even under drastic sparsification/quantization, and consequently that very little downstream performance loss occurs when working with very aggressively sparsified or quantized spectral clustering problems. We illustrate how these results depend on the nonlinearity, we characterize a phase transition beyond which spectral clustering becomes possible, and we show when such nonlinear transformations can introduce spurious non-informative eigenvectors.

## 1 INTRODUCTION

Sparsifying, quantizing, and/or performing other entry-wise nonlinear operations on large matrices can have many benefits. Historically, this has been used to develop iterative algorithms for core numerical linear algebra problems (Achlioptas & McSherry, 2007; Drineas & Zouzias, 2011). More recently, this has been used to design better neural network models (Srivastava et al., 2014; Dong et al., 2019; Shen et al., 2020). A concrete example, amenable to theoretical analysis and ubiquitous in practice, is provided by spectral clustering, which can be solved by retrieving the dominant eigenvectors of $\mathbf{X}^\mathsf{T}\mathbf{X}$, for $\mathbf{X} = [\mathbf{x}_1, \ldots, \mathbf{x}_n] \in \mathbb{R}^{p \times n}$ a large data matrix (Von Luxburg, 2007). When the amount of data $n$ is large, the Gram "kernel" matrix $\mathbf{X}^\mathsf{T}\mathbf{X}$ can be enormous, impractical even to form and leading to computationally unaffordable algorithms. For instance, Lanczos iteration that operates through repeated matrix-vector multiplication suffers from an $O(n^2)$ complexity (Golub & Loan, 2013) and quickly becomes burdensome.

One approach to overcoming this limitation is simple subsampling: dividing $\mathbf{X}$ into subsamples of size $\varepsilon n$, for some $\varepsilon \in (0, 1)$, on which one performs parallel computation, and then recombining. This leads to computational gain, but at the cost of degraded performance, since each data point $\mathbf{x}_i$ looses the *cumulative effect* of comparing to the *whole* dataset. An alternative cost-reduction procedure consists in *uniformly* randomly "zeroing-out" entries from the whole matrix $\mathbf{X}^\mathsf{T}\mathbf{X}$, resulting in a sparse matrix with only an $\varepsilon$ fraction of nonzero entries. For spectral clustering, by focusing on the eigenspectrum of the "zeroed-out" matrix, Zarrouk et al. (2020) showed that the same computational gain can be achieved at the cost of a much less degraded performance: for $n/p$ rather large, almost no degradation is observed down to very small values of $\varepsilon$ (e.g., $\varepsilon \approx 2\%$ for $n/p \gtrsim 100$).

Previous efforts showed that it is often advantageous to perform sparsification/quantization in a *non-uniform* manner, rather than uniformly (Achlioptas & McSherry, 2007; Drineas & Zouzias, 2011). The focus there, however, is often on (non-asymptotic bounds of) the approximation error between

the original and the sparsified/quantized matrices. This, however, does *not* provide a direct access to the actual performance for spectral clustering or other downstream tasks of interest, e.g., since the top eigenvectors are known to exhibit a phase transition phenomenon (Baik et al., 2005; Saade et al., 2014). That is, they can behave very differently from those of the original matrix, even if the matrix after treatment is close in operator or Frobenius norm to the original matrix.

Here, we focus on a *precise* characterization of the eigenstructure of $\mathbf{X}^\mathsf{T}\mathbf{X}$ after entry-wise nonlinear transformation such as sparsification or quantization, in the large $n, p$ regime, by performing simultaneously *non-uniform sparsification and/or quantization* (down to binarization). We consider a simple mixture data model with $\mathbf{x} \sim \mathcal{N}(\pm\boldsymbol{\mu}, \mathbf{I}_p)$ and let $\mathbf{K} \equiv f(\mathbf{X}^\mathsf{T}\mathbf{X}/\sqrt{p})/\sqrt{p}$, where $f$ is an entry-wise thresholding/quantization operator (thereby zeroing-out/quantizing entries of $\mathbf{X}^\mathsf{T}\mathbf{X}$); and we prove that this leads to significantly improved performances, with the same computational cost, in spectral clustering as uniform sparsification, but for a much reduced cost in storage induced by quantization. The only (non-negligible) additional cost arises from the extra need for evaluating each entry of $\mathbf{X}^\mathsf{T}\mathbf{X}$. Our main technical contribution (of independent interest, e.g., for those interested in entry-wise nonlinear transformations of feature matrices) consists in using random matrix theory (RMT) to derive the large $n, p$ asymptotics of the eigenspectrum of $\mathbf{K} = f(\mathbf{X}^\mathsf{T}\mathbf{X}/\sqrt{p})/\sqrt{p}$ for a wide range of functions $f$, and then comparing to previously-established results for uniform subsampling and sparsification in (Zarrouk et al., 2020). Experiments on real-world data further corroborate our findings.

Our main contributions are the following.

1. We derive the limiting eigenvalue distribution of $\mathbf{K}$ as $n, p \to \infty$ (Theorem 1), and we identify:
   (a) the existence of *non-informative* and *isolated* eigenvectors of $\mathbf{K}$ for some $f$ (Corollary 1);
   (b) in the absence of such eigenvectors, a phase transition in the dominant eigenvalue-eigenvector $(\hat{\lambda}, \hat{\mathbf{v}})$ pair (Corollary 2): if the signal-to-noise ratio (SNR) $\|\boldsymbol{\mu}\|^2$ of the data exceeds a certain threshold $\gamma$, then $\hat{\lambda}$ becomes isolated from the main bulk (Von Luxburg, 2007; Joseph & Yu, 2016; Baik et al., 2005) and $\hat{\mathbf{v}}$ contains data class-structure information exploitable for clustering; if not, then $\hat{\mathbf{v}}$ contains only noise and is asymptotically *orthogonal* to the class-label vector.

2. Letting $f$ be a sparsification, quantization, or binarization operator, we show:
   (a) a *selective* non-uniform sparsification operator, such that $\mathbf{X}^\mathsf{T}\mathbf{X}$ can be drastically sparsified with very little degradation in clustering performance (Proposition 1 and Section 4.2), which significantly outperforms the random uniform sparsification scheme in (Zarrouk et al., 2020);
   (b) for a given matrix storage budget (i.e., fixed number of bits to store $\mathbf{K}$), an optimal design of the quantization/binarization operators (Proposition 2 and Section 4.3), the performances of which are compared against the original $\mathbf{X}^\mathsf{T}\mathbf{X}$ and its sparsified but not quantized version.

For spectral clustering, the surprisingly small performance drop, accompanied by a huge reduction in computational cost, contributes to improved algorithms for large-scale problems. More generally, our proposed analysis sheds light on the effect of entry-wise nonlinear transformations on the eigenspectra of data/feature matrices. Thus, looking forward (and perhaps more importantly, given the use of nonlinear transformations in designing modern neural network models as well as the recent interest in applying RMT to neural network analyses (Dobriban et al., 2018; Li & Nguyen, 2018; Seddik et al., 2018; Jacot et al., 2019; Liu & Dobriban, 2019)), we expect that our analysis opens the door to improved analysis of computationally efficient methods for large dimensional machine learning and neural network models more generally.

## 2 SYSTEM MODEL AND PRELIMINARIES

**Basic setup.** Let $\mathbf{x}_1, \ldots, \mathbf{x}_n \in \mathbb{R}^p$ be independently drawn (not necessarily uniformly) from a two-class mixture of $\mathcal{C}_1$ and $\mathcal{C}_2$ with

$$\mathcal{C}_1 : \mathbf{x}_i = -\boldsymbol{\mu} + \mathbf{z}_i, \quad \mathcal{C}_2 : \mathbf{x}_i = +\boldsymbol{\mu} + \mathbf{z}_i \tag{1}$$

with $\mathbf{z}_i \in \mathbb{R}^p$ having i.i.d. zero-mean, unit-variance, $\kappa$-kurtosis, sub-exponential entries, $\boldsymbol{\mu} \in \mathbb{R}^p$ such that $\|\boldsymbol{\mu}\|^2 \to \rho \geq 0$ as $p \to \infty$, and $\mathbf{v} \in \{\pm 1\}^n$ with $[\mathbf{v}]_i = -1$ for $\mathbf{x}_i \in \mathcal{C}_1$ and $+1$ for $\mathbf{x}_i \in \mathcal{C}_2$.[1] The data matrix $\mathbf{X} = [\mathbf{x}_1, \ldots, \mathbf{x}_n] \in \mathbb{R}^{p \times n}$ can be compactly written as $\mathbf{X} = \mathbf{Z} + \boldsymbol{\mu}\mathbf{v}^\mathsf{T}$

---

[1]The norm $\|\cdot\|$ is the Euclidean norm for vectors and the operator norm for matrices.

for $\mathbf{Z} = [\mathbf{z}_1, \ldots, \mathbf{z}_n]$ so that $\|\mathbf{v}\| = \sqrt{n}$ and both $\mathbf{Z}, \boldsymbol{\mu}\mathbf{v}^\mathsf{T}$ have operator norm of order $O(\sqrt{p})$ in the $n \sim p$ regime. In this setting, the Gram (or *linear* kernel) matrix $\mathbf{X}^\mathsf{T}\mathbf{X}$ achieves optimal clustering performance on the mixture model (1); see Remark 1 below. However, it consists of a dense $n \times n$ matrix, which becomes quickly expensive to store or to perform computation on, as $n$ increases.

Thus, we consider instead the following *entry-wise nonlinear* transformation of $\mathbf{X}^\mathsf{T}\mathbf{X}$:

$$\mathbf{K} = \left\{ \delta_{i \neq j} f(\mathbf{x}_i^\mathsf{T} \mathbf{x}_j / \sqrt{p}) / \sqrt{p} \right\}_{i,j=1}^n \tag{2}$$

for $f : \mathbb{R} \to \mathbb{R}$ satisfying some regularity conditions (see Assumption 1 below), where $\delta_{i \neq j}$ equals 1 for $i \neq j$ and equals 0 otherwise. The diagonal elements $f(\mathbf{x}_i^\mathsf{T} \mathbf{x}_i / \sqrt{p})$ (**i**) bring no additional information for clustering and (**ii**) do not scale properly for $p$ large ($\mathbf{x}_i^\mathsf{T} \mathbf{x}_i / \sqrt{p} = O(\sqrt{p})$ instead of $O(1)$). Thus, following (El Karoui, 2010; Cheng & Singer, 2013), they are discarded.

Most of our technical results hold for rather generic functions $f$, e.g., those of interest beyond sparse quantized spectral clustering, but we are particularly interested in $f$ with nontrivial numerical properties (e.g., promoting quantization and sparsity):

$$\textbf{Sparsification:} \qquad f_1(t) = t \cdot 1_{|t| > \sqrt{2}s} \tag{3}$$

$$\textbf{Quantization:} \qquad f_2(t) = 2^{2-M}(\lfloor t \cdot 2^{M-2} / \sqrt{2}s \rfloor + 1/2) \cdot 1_{|t| \leq \sqrt{2}s} + \text{sign}(t) \cdot 1_{|t| > \sqrt{2}s} \tag{4}$$

$$\textbf{Binarization:} \qquad f_3(t) = \text{sign}(t) \cdot 1_{|t| > \sqrt{2}s} \cdot \tag{5}$$

Here, $s \geq 0$ is some *truncation threshold*, and $M \geq 2$ is a number of information bits.[2] The visual representations of these $f$s are given in Figure 1-(left). For $f_3$, taking $s \to 0$ leads to the sign function $\text{sign}(t)$. In terms of storage, the quantization $f_2$ consumes $2^{M-2} + 1$ bits per nonzero entry, while the binarization $f_3$ takes values in $\{\pm 1, 0\}$ and thus consumes 1 bit per nonzero entry.

**Random matrix theory.** To provide a precise description of the eigenspectrum of $\mathbf{K}$ for the nonlinear $f$ of interest, to be used in the context of spectral clustering, we will provide a large dimensional characterization for the *resolvent* of $\mathbf{K}$, defined for $z \in \mathbb{C} \setminus \mathbb{R}^+$ as

$$\mathbf{Q}(z) \equiv (\mathbf{K} - z\mathbf{I}_n)^{-1}. \tag{6}$$

This matrix, which plays a central role in RMT analysis (Bai & Silverstein, 2010), will be used in two primary ways. First, the normalized trace $\frac{1}{n} \text{tr} \, \mathbf{Q}(z)$ is the so-called *Stieltjes transform* of the eigenvalue distribution of $\mathbf{K}$, from which the eigenvalue distribution can be recovered, and be further used to characterize the phase transition beyond which spectral clustering becomes theoretically possible (Corollary 2). Second, for $(\hat{\lambda}, \hat{\mathbf{v}})$, an "isolated" eigenvalue-eigenvector pair of $\mathbf{K}$, and $\mathbf{a} \in \mathbb{R}^n$, a deterministic vector, by Cauchy's integral formula, the "angle" between $\hat{\mathbf{v}}$ and $\mathbf{a}$ is given by $|\hat{\mathbf{v}}^\mathsf{T}\mathbf{a}|^2 = -\frac{1}{2\pi i} \oint_{\Gamma(\hat{\lambda})} \mathbf{a}^\mathsf{T}\mathbf{Q}(z)\mathbf{a} \, dz$, where $\Gamma(\hat{\lambda})$ is a positively oriented contour surrounding $\hat{\lambda}$ only. Letting $\mathbf{a} = \mathbf{v}$, this will be exploited to characterize the spectral clustering error rate (Proposition 1).

From a technical perspective, unlike linear random matrix models, $\mathbf{K}$ (and thus $\mathbf{Q}(z)$) involves nonlinear dependence between its entries. To break this difficulty, following the ideas of (Cheng & Singer, 2013), we exploit the fact that, by the central limit theorem, $\mathbf{z}_i^\mathsf{T}\mathbf{z}_j / \sqrt{p} \to \mathcal{N}(0, 1)$ in distribution as $p \to \infty$. As such, up to $\boldsymbol{\mu}\mathbf{v}^\mathsf{T}$, which is treated separately with a perturbation argument, the entries of $\mathbf{K}$ asymptotically behave like a family of *dependent* standard Gaussian variables to which $f$ is applied. Expanding $f$ in a series of *orthogonal polynomials* with respect to the Gaussian measure allows for "unwrapping" this dependence. A few words on the theory of orthogonal polynomials (Andrews et al., 1999) are thus convenient to pursue our analysis.

**Orthogonal polynomial framework.** For a probability measure $\mu$, let $\{P_l(x), l \geq 0\}$ be the orthonormal polynomials with respect to $\langle f, g \rangle \equiv \int fg d\mu$ obtained by the Gram-Schmidt procedure on the monomials $\{1, x, x^2, \ldots\}$, such that $P_0(x) = 1$, $P_l$ is of degree $l$ and $\langle P_{l_1}, P_{l_2} \rangle = \delta_{l_1 - l_2}$. By the Riesz-Fischer theorem (Rudin, 1964, Theorem 11.43), for any function $f \in L^2(\mu)$, the set of square-integrable functions with respect to $\langle \cdot, \cdot \rangle$, one can formally expand $f$ as

$$f(x) \sim \sum_{l=0}^\infty a_l P_l(x), \quad a_l = \int f(x) P_l(x) \mu(dx) \tag{7}$$

---

[2]Also, here, $\lfloor \cdot \rfloor$ denotes the floor function, while $\text{erf}(x) = \frac{2}{\sqrt{\pi}} \int_0^x e^{-t^2} dt$ and $\text{erfc}(x) = 1 - \text{erf}(x)$ denotes the error function and complementary error function, respectively.

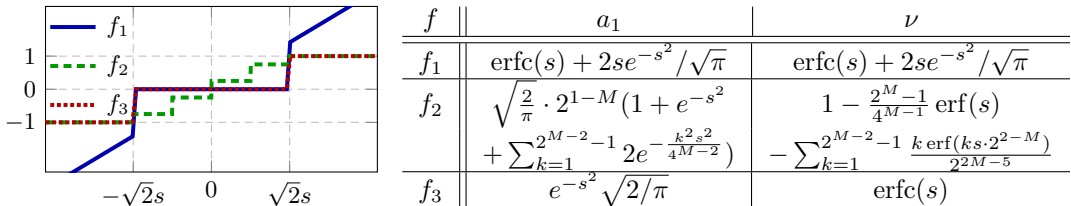

Figure 1: Visual representations of different functions $f$ defined in (3)-(5) **(left)** and their corresponding parameters $a_1$ and $\nu$ **(right)** ($a_2 = 0$ for each of these $f$s) defined in Assumption 1.

where "$f \sim \sum_{l=0}^{\infty} a_l P_l$" indicates that $\|f - \sum_{l=0}^{L} a_l P_l\| \to 0$ as $L \to \infty$ with $\|f\|^2 = \langle f, f \rangle$.

To investigate the asymptotic behavior of $\mathbf{K}$, as $n, p \to \infty$, we make the following assumption on $f$.

**Assumption 1** (Square-integrable in Gauss space). *Let $\xi_p = \mathbf{z}_i^\mathsf{T} \mathbf{z}_j / \sqrt{p}$ and $P_{l,p}(x)$ be the orthonormal polynomials with respect to the measure $\mu_p$ of $\xi_p$. For $f \in L^2(\mu_p)$, $f(x) \sim \sum_{l=0}^{\infty} a_{l,p} P_{l,p}(x)$ with $a_{l,p}$ in (7) such that (i) $\sum_{l=0}^{\infty} a_{l,p} P_{l,p}(x) \mu_p(dx)$ converges in $L^2(\mu_p)$ to $f(x)$ uniformly over large $p$; and (ii) as $p \to \infty$, $\sum_{l=1}^{\infty} a_{l,p}^2 \to \nu$ and, for $l = 0, 1, 2$, $a_{l,p} \to a_l$ converges with $a_0 = 0$.*

Since $\xi_p \to \mathcal{N}(0, 1)$ in distribution, the parameters $a_0, a_1, a_2$ and $\nu$ are simply moments of the standard Gaussian measure involving $f$. More precisely, for $\xi \sim \mathcal{N}(0, 1)$,

$$a_0 = \mathbb{E}[f(\xi)], \quad a_1 = \mathbb{E}[\xi f(\xi)], \quad \sqrt{2} a_2 = \mathbb{E}[\xi^2 f(\xi)] - a_0, \quad \nu = \mathbb{E}[f^2(\xi) - a_0^2] \geq a_1^2 + a_2^2. \quad (8)$$

Imposing the condition $a_0 = 0$ simply discards the *non-informative* rank-one matrix $a_0 \mathbf{1}_n \mathbf{1}_n^\mathsf{T} / \sqrt{p}$ from $\mathbf{K}$. The three parameters $(a_1, a_2, \nu)$ are of crucial significance in determining the spectral behavior of $\mathbf{K}$ (see Theorem 1 below). The sparse $f_1$, quantized $f_2$, and binary $f_3$ of our primary interest *all* satisfy Assumption 1 (counterexample exists though, for example $f(x) = e^{P(x)}$ for polynomial $P(x)$ of degree greater than two), with their corresponding $a_2 = 0$ (as we shall see in Corollary 1 below, this is important for the spectral clustering use case) and $a_1, \nu$ given in Figure 1-(right). With these preliminary comments, we are in position to present our main technical results.

## 3 MAIN TECHNICAL RESULTS

Our main technical result, from which our performance-complexity trade-off analysis will follow, provides an asymptotic *deterministic equivalent* $\bar{\mathbf{Q}}(z)$ for the *random resolvent* $\mathbf{Q}$, defined in (6). (A deterministic equivalent is a deterministic matrix $\bar{\mathbf{Q}}(z)$ such that, for any deterministic sequence of matrices $\mathbf{A}_n \in \mathbb{R}^{n \times n}$ and vectors $\mathbf{a}_n, \mathbf{b}_n \in \mathbb{R}^n$ of bounded (spectral and Euclidean) norms, $\frac{1}{n} \operatorname{tr} \mathbf{A}_n(\mathbf{Q}(z) - \bar{\mathbf{Q}}(z)) \to 0$ and $\mathbf{a}_n^\mathsf{T}(\mathbf{Q}(z) - \bar{\mathbf{Q}}(z))\mathbf{b}_n \to 0$ almost surely as $n, p \to \infty$. We denote this relation $\mathbf{Q}(z) \leftrightarrow \bar{\mathbf{Q}}(z)$.) This is given in the following theorem. The proof is in Appendix A.1.

**Theorem 1** (Deterministic equivalent). *Let $n, p \to \infty$ with $p/n \to c \in (0, \infty)$, let $\mathbf{Q}(z)$ be defined in (6), and let $\Im[\cdot]$ denote the imaginary part of a complex number. Then, under Assumption 1,*

$$\mathbf{Q}(z) \leftrightarrow \bar{\mathbf{Q}}(z) = m(z)\mathbf{I}_n - \mathbf{V}\mathbf{\Lambda}(z)\mathbf{V}^\mathsf{T}, \quad \mathbf{\Lambda}(z) = \begin{bmatrix} \Theta(z)m^2(z) & \frac{\mathbf{v}^\mathsf{T}\mathbf{1}_n \Theta(z)\Omega(z)}{n} m(z) \\ \frac{\mathbf{v}^\mathsf{T}\mathbf{1}_n \Theta(z)\Omega(z)}{n} m(z) & \frac{(\mathbf{v}^\mathsf{T}\mathbf{1}_n)^2 \Theta(z)\Omega^2(z)}{n^2} - \Omega(z) \end{bmatrix},$$

*with $\sqrt{n}\mathbf{V} = [\mathbf{v}, \mathbf{1}_n]$, $\Omega(z) = \frac{a_2^2(\kappa-1)m^3(z)}{2c^2 - a_2^2(\kappa-1)m^2(z)}$, $\Theta(z) = \frac{a_1\|\boldsymbol{\mu}\|^2}{c + a_1 m(z)(1+\|\boldsymbol{\mu}\|^2) + a_1\|\boldsymbol{\mu}\|^2 \Omega(z)(\mathbf{v}^\mathsf{T}\mathbf{1}_n)^2/n^2}$, for $\kappa$ the kurtosis of the entries of $\mathbf{Z}$, and $m(z)$ the unique solution, such that $\Im[m(z)] \cdot \Im[z] \geq 0$, to*

$$-\frac{1}{m(z)} = z + \frac{a_1^2 m(z)}{c + a_1 m(z)} + \frac{\nu - a_1^2}{c} m(z). \quad (9)$$

The major implication of Theorem 1 is that, the spectral behavior of the matrix $\mathbf{K}$ (e.g., its eigenvalue distribution and the isolated eigenpairs as discussed after (6)) depends on the nonlinear $f$ only via the *three* parameters $a_1, a_2, \nu$ defined in (8). More concretely, the *empirical spectral measure* $\omega_n = \frac{1}{n} \sum_{i=1}^{n} \delta_{\lambda_i(\mathbf{K})}$ of $\mathbf{K}$ has a deterministic limit $\omega$ as $n, p \to \infty$, uniquely defined through its Stieltjes transform $m(z) \equiv \int (t - z)^{-1} \omega(dt)$ as the solution to (9).[3] This limiting measure $\omega$ does *not*

---

[3]For $m(z)$ the Stieltjes transform of a measure $\omega$, $\omega$ can be obtained via $\omega([a, b]) = \lim_{y \downarrow 0} \frac{1}{\pi} \int_a^b \Im[m(x + \imath y)] dx$ for all $a < b$ continuity points of $\omega$.

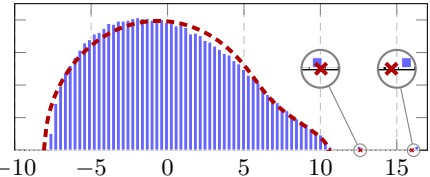 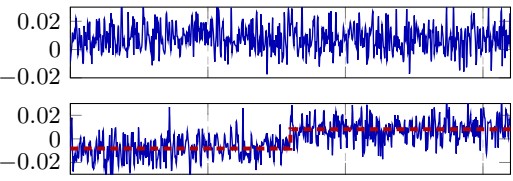

Figure 2: **(Left)** Histogram of eigenvalues of $\mathbf{K}$ (**blue**) versus the limiting spectrum and spikes (**red**). **(Right)** Eigenvectors of the largest **(top)** and second largest **(bottom)** eigenvalues of $\mathbf{K}$ (**blue**), versus the rescaled class label $\alpha\mathbf{v}/\sqrt{n}$ (**red**, from Corollary 2). $f(t) = \sin(t) - 3\cos(t) + 3/\sqrt{e}$, $p = 800$, $n = 6\,400$, $\boldsymbol{\mu} = 1.1 \cdot \mathbf{1}_p/\sqrt{p}$, $\mathbf{v} = [-\mathbf{1}_{n/2}; \mathbf{1}_{n/2}]$ on Student-t data with $\kappa = 5$.

depend on the law of the independent entries of $\mathbf{Z}$, so long that they are sub-exponential, with zero mean and unit variance. In particular, taking $a_1 = 0$ in (9) gives the (rescaled) Wigner semi-circle law $\omega$ (Wigner, 1955), and taking $\nu = a_1^2$ (i.e., $a_l = 0$ for $l \geq 2$) gives the Marčenko-Pastur law $\omega$ (Marcenko & Pastur, 1967). See Remark 2 in Appendix A.2 for more discussions.

**Spurious non-informative spikes.** Going beyond just the limiting spectral measure of $\mathbf{K}$, Theorem 1 also shows that *isolated eigenvalues* (often referred to as *spikes*) may be found in the eigenspectrum of $\mathbf{K}$ at very specific locations. Such spikes and their associated eigenvectors are typically thought to provide information on the data class-structure (and they do when $f$ is linear). However, when $f$ is nonlinear, this is not always the case: it is possible that *not all* these spikes are "useful" in the sense of being informative about the class structure. This is made precise in the following, where we assume $\boldsymbol{\mu} = \mathbf{0}$, i.e., that there is no class structure. The proof is in Appendix A.2.

**Corollary 1** (Non-informative spikes). *Assume $\boldsymbol{\mu} = \mathbf{0}$, $\kappa \neq 1$ and $a_2 \neq 0$, so that in the notations of Theorem 1, $\Theta(z) = 0$ and $\bar{\mathbf{Q}}(z) = m(z) + \Omega(z)\frac{1}{n}\mathbf{1}_n\mathbf{1}_n^\mathsf{T}$ for $\Omega(z)$ defined in Theorem 1. Then, if $x_\pm \equiv \pm\frac{1}{a_2}\sqrt{\frac{2}{\kappa-1}}$ satisfies $a_1 x_\pm \neq \pm 1$ and $(\nu - a_1^2)x_\pm^2 + \frac{a_1^2 x_\pm^2}{(1+a_1 x_\pm)^2} < 1/c$, two eigenvalues of $\mathbf{K}$ converge to $z_\pm = -\frac{1}{cx_\pm} - \frac{a_1^2 x_\pm}{1+a_1 x_\pm} - (\nu - a_1^2)x_\pm$ away from the support of $\omega$. If instead $x_\pm = \pm 1/a_1$ for $a_1 \neq 0$, then a single eigenvalue of $\mathbf{K}$ isolates with limit $z = -\frac{\nu}{a_1} - \frac{a_1(2-c)}{2c}$.*

Corollary 1 (combined with Theorem 1) says that, while the limiting spectrum $\omega$ is *universal* with respect to the distribution of the entries of $\mathbf{Z}$, the existence and position of the spurious non-informative spikes are *not universal* and depend on the kurtosis $\kappa$ of the distribution. (See Figure 12 in Appendix A.2 for an example.) This is far from a mathematical curiosity. Given how nonlinear transformations are used in machine learning practice, being aware of the existence of *spurious non-informative spikes* in the eigenspectrum of $\mathbf{K}$ (well separated from the bulk of eigenvalues, but that correspond to random noise instead of signal), as a function of properties of the nonlinear $f$, is of fundamental importance for downstream tasks. For example, for spectral clustering, their associated eigenvectors may be mistaken as informative ones by spectral clustering algorithms, even in the complete absence of classes. This is confirmed by Figure 2 where two isolated eigenvalues (on the right side of the bulk) are observed, with only the *second largest* one corresponding to an eigenvector that contains class-label information. For further discussions, see Appendix A.2 and A.3.

**Informative spikes.** From Theorem 1, we see that the eigenspectrum of $\mathbf{K}$ depends on $f$ only via $a_1$, $a_2$, and $\nu$. In particular, $a_1$ and $\nu$ determine the limiting spectral measure $\omega$. From Corollary 1, we see that $a_2$ contributes by (**i**) introducing (at most two) non-informative spikes and (**ii**) reducing the ratio $a_1/\nu$ (since $\nu = \sum_{i\geq 1} a_i^2$), thereby necessarily *enlarging* the support of $\omega$ (see Remark 2 in Appendix A.2 for more details). Taking $a_2 = 0$ thus reduces the length of the support of $\omega$ and, as such, maximizes the "chance" of the appearance of an informative spike (the eigenvector of which is positively correlated with the label vector $\mathbf{v}$). See Remark 1 below for a more precise statement.

In particular, by taking $a_2 = 0$ and $a_1 \neq 0$, we obtain only informative spikes, and we can characterize a phase transition depending on the SNR $\rho$. The proof of the following is in Appendix A.3.

**Corollary 2** (Informative spike and a phase transition). *For $a_1 > 0$ and $a_2 = 0$, let*

$$F(x) = x^4 + 2x^3 + \left(1 - \frac{c\nu}{a_1^2}\right)x^2 - 2cx - c, \quad G(x) = \frac{a_1}{c}(1+x) + \frac{a_1}{x} + \frac{\nu - a_1^2}{a_1}\frac{1}{1+x}, \quad (10)$$

*and let $\gamma$ be the largest real solution to $F(\gamma) = 0$. Then, under Assumption 1, we have $\sqrt{c} \leq \gamma \leq \sqrt{c\nu}/a_1$, and the largest eigenpair $(\hat{\lambda}, \hat{\mathbf{v}})$ of $\mathbf{K}$ satisfies*

$$\hat{\lambda} \to \lambda = \begin{cases} G(\rho) & \rho > \gamma, \\ G(\gamma) & \rho \leq \gamma; \end{cases} \qquad \frac{|\hat{\mathbf{v}}^{\mathsf{T}}\mathbf{v}|^2}{n} \to \alpha = \begin{cases} \frac{F(\rho)}{\rho(1+\rho)^3} & \rho > \gamma, \\ 0 & \rho \leq \gamma; \end{cases} \qquad (11)$$

*almost surely as $n, p \to \infty$, $p/n \to c \in (0, \infty)$, where we recall $\rho \equiv \lim_{p\to\infty} \|\boldsymbol{\mu}\|^2$ and $\|\mathbf{v}\| = \sqrt{n}$.*

Without loss of generality, we discuss only the case $a_1 > 0.$[4] For $a_1 > 0$, both $F(x)$ and $G(x)$ are increasing functions on $x \in (\gamma, \infty)$. Then, as expected, both $\lambda$ and $\alpha$ increase with the SNR $\rho$. Moreover, the phase transition point $\gamma$ is an increasing function of $c$ and of $\nu/a_1^2$. As such, the optimal $f$ in the sense of the smallest phase transition point is the linear function $f(t) = t$ with $a_1 = \nu = 1$ and $\gamma = \sqrt{c}$. This recovers the classical random matrix result in (Baik et al., 2005).

## 4    CLUSTERING PERFORMANCE OF SPARSE AND QUANTIZED OPERATORS

We start, in Section 4.1, by providing a sharp asymptotic characterization of the clustering error rate and demonstrating the optimality of the linear function under (1). Then, in Section 4.2, we discuss the advantageous performance of the proposed *selective* sparsification approach (with $f_1$) versus the uniform or subsampling approach studied previously in (Zarrouk et al., 2020). Finally, in Section 4.3, we derive the optimal truncation threshold $s_{\mathrm{opt}}$, for both quantized $f_2$ and binary $f_3$, so as to achieve an optimal performance-complexity trade-off for a given storage budget.

### 4.1    PERFORMANCE OF SPECTRAL CLUSTERING

The technical results in Section 3 provide conditions under which $\mathbf{K}$ admits an informative eigenvector $\hat{\mathbf{v}}$ that is non-trivially correlated with the class label vector $\mathbf{v}$ (and thus that is exploitable for spectral clustering) in the $n, p \to \infty$ limit. Since the exact (limiting) alignment $|\mathbf{v}^{\mathsf{T}}\hat{\mathbf{v}}|$ is known, along with an additional argument on the normal fluctuations of $\hat{\mathbf{v}}$, we have the following result for the performance of the spectral clustering method. The proof is in Appendix A.4.

**Proposition 1** (Performance of spectral clustering). *Let Assumption 1 hold, let $a_1 > 0, a_2 = 0$, and let $\hat{\mathcal{C}}_i = \mathrm{sign}([\hat{\mathbf{v}}]_i)$ be the estimate of the underlying class $\mathcal{C}_i$ of $\mathbf{x}_i$, with the convention $\hat{\mathbf{v}}^{\mathsf{T}}\mathbf{v} \geq 0$, for $\hat{\mathbf{v}}$ the top eigenvector of $\mathbf{K}$. Then, the (average) misclassification rate satisfies $\frac{1}{n}\sum_{i=1}^{n} \delta_{\hat{\mathcal{C}}_i \neq \mathcal{C}_i} \to \frac{1}{2}\mathrm{erfc}(\sqrt{\alpha/(2-2\alpha)})$, almost surely, as $n, p \to \infty$, for $\alpha \in [0, 1)$ defined in (11).*

Recall from Corollary 2 that, for $a_2 = 0$, the nonlinear function $f$ (e.g., $f_1, f_2, f_3$) acts on the (statistical behavior of the) isolated eigenvector $\hat{\mathbf{v}}$, and thus on the spectral clustering performance per Proposition 1, *only* through the ratio $\nu/a_1^2$. It thus suffices to evaluate the ratio $\nu/a_1^2$ of different $f$ and compare, for instance to that of the linear $f(t) = t$ corresponding to the original $\mathbf{X}^{\mathsf{T}}\mathbf{X}$ matrix.

Despite being asymptotic results valid in the $n, p \to \infty$ limit, the results of Proposition 1 and Corollary 2 closely match empirical results for moderately large $n, p$ only in the hundreds. This is illustrated in Figure 3. Proposition 1 further confirms that the misclassification rate, being a decreasing function of $\alpha$, increases with $\nu/a_1^2$ (for $c$ and $\rho$ fixed). This leads to the following remark.

**Remark 1** (Optimality of linear function). *Since both the phase transition point $\gamma$ and the misclassification rate grow with $\nu/a_1^2$, the linear function $f(t) = t$ with the minimal $\nu/a_1^2 = 1$ is* optimal *in the sense of (**i**) achieving the* smallest *SNR $\rho$ or the* largest *ratio $c = \lim p/n$ (i.e., the fewest samples $n$) necessary to observe an informative (isolated) eigenvector, and (**ii**) upon existence of such an isolated eigenvector, reaching the* lowest *classification error rate.*

According to Remark 1, any $f$ with $\nu/a_1^2 > 1$ induces performance degeneration (compared to the optimal linear function). However, by choosing $f$ to be one of the functions $f_1, f_2, f_3$ defined in (3)-(5), one may trade clustering performance optimality for reduced storage size and computational time. Figure 4 displays the theoretical decay in clustering performance and the gain in storage size of the sparse $f_1$, quantized $f_2$, and binary $f_3$, when compared to the optimal but "dense" linear function. As $s \to 0$, both performance and storage size under $f_1$ naturally approach those of the

---

[4]Otherwise we could consider $-\mathbf{K}$ instead of $\mathbf{K}$ and the largest eigenvalue becomes the smallest one.

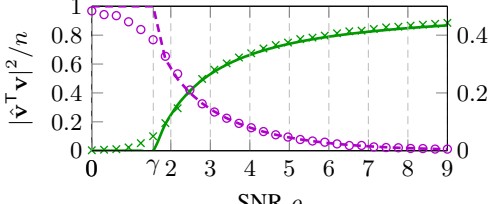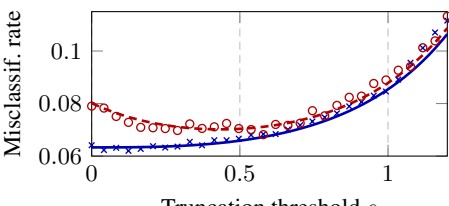

Figure 3: **(Left)** Empirical alignment $|\hat{\mathbf{v}}^{\mathsf{T}}\mathbf{v}|^2/n$ (**green crosses**) and misclassification rate (**purple circles**) in markers versus their limiting behaviors in lines, for $f(t) = \text{sign}(t)$, as a function of SNR $\rho$. **(Right)** Misclassification rate as a function of the truncation thresholds $s$ of sparse $f_1$ (**blue crosses**) and binary $f_3$ (**red circles**) with $\rho = 4$. Here, $p = 512$, $n = 256$, $\boldsymbol{\mu} \propto \mathcal{N}(\mathbf{0}, \mathbf{I}_p)$, $\mathbf{v} = [-\mathbf{1}_{n/2}; \mathbf{1}_{n/2}]$ on Gaussian data. The results are obtained by averaging over 250 runs.

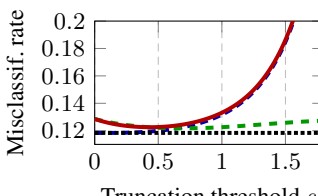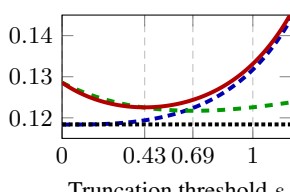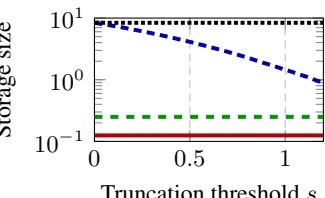

Figure 4: Clustering performance (**left**, a zoom-in in **middle**) and storage size (MB) (**right**) of $f_1$ (**blue**), $f_2$ with $M = 2$ (**green**), $f_3$ (**red**), and linear $f(t) = t$ (**black**), versus the truncation threshold $s$, for SNR $\rho = 2$, $c = 1/2$ and $n = 10^3$, with 64 bits per entry for non-quantized matrices.

linear function. This is unlike $f_2$ or $f_3$, which approach the sign function. For $s \gg 1$, the performance under sparse $f_1$ becomes comparable to that of binary $f_3$ (which is significantly worse than quantized but non-sparse $f_2$) but for a larger storage cost. In particular, using $f_2$ or $f_3$ in the setting of Figure 4, one can reduce the storage size by a factor of 32 or 64 (in the case of IEEE standard single- or double-precision floating-point format), at the price of a performance drop less than 1%.

## 4.2 COMPARISON TO UNIFORM SPARSIFICATION AND SUBSAMPLING

From Figure 4, we see that the classification error and storage gain of the sparse $f_1$ increase monotonically, as the truncation threshold $s$ grows. For $f_1$, the number of nonzero entries of $\mathbf{K}$ is approximately $\text{erfc}(s)n^2$ with truncation threshold $s$. Thus, the *sparsity level* $\varepsilon_{\text{selec}} = \text{erfc}(s) \in [0, 1]$ can be defined and compared to uniform sparsification or subsampling approaches.

Recall (from the introduction) that the cost of spectral clustering may be reduced by subsampling the whole dataset in $1/\varepsilon_{\text{sub}}$ chunks of $n\varepsilon_{\text{sub}}$ data vectors each. Alternatively, as investigated recently in (Zarrouk et al., 2020), the cost can be reduced by uniformly zeroing-out $\mathbf{X}^{\mathsf{T}}\mathbf{X}$ with a symmetric random mask matrix $\mathbf{B}$, with $\mathbf{B}_{ij} \sim \text{Bern}(\varepsilon_{\text{unif}})$ for $1 \leq i < j \leq n$ and $\mathbf{B}_{ii} = 0$. On average, a proportion $1 - \varepsilon_{\text{unif}}$ of the entries of $\mathbf{X}^{\mathsf{T}}\mathbf{X}$ is set to zero, so that $\varepsilon_{\text{unif}} \in [0, 1]$ controls the sparsity level (and thus the storage size as well as computational time). Similar to our Corollary 2, the associated eigenvector alignment $\alpha$ (and thus the clustering accuracy per Proposition 1) in both cases can be derived. Specifically, taking $\varepsilon_{\text{unif}} = a_1^2/\nu$ in (Zarrouk et al., 2020, Theorem 3.2), we obtain the same $F(x)$ as in our Corollary 2 and therefore the same phase transition point $\gamma$ and eigenvector alignment $\alpha$. As for subsampling, its performance can be obtained by letting $a_1^2 = \nu$ and changing $c$ into $c/\varepsilon_{\text{sub}}$ in the formulas of $F(x)$ and $G(x)$ of our Corollary 2. Consequently, the same clustering performance is achieved by either uniform or selective sparsification (using $f_1$) with

$$\varepsilon_{\text{unif}} = a_1^2/\nu = \text{erfc}(s) + 2se^{-s^2}/\sqrt{\pi} > \text{erfc}(s) = \varepsilon_{\text{selec}}, \tag{12}$$

and our proposed selective sparsification thus leads to *strictly sparser* matrices. Moreover, their ratio $r(s) = \text{erfc}(s)/(\text{erfc}(s) + 2se^{-s^2}/\sqrt{\pi})$ is a decreasing function of $s$ and approximates as

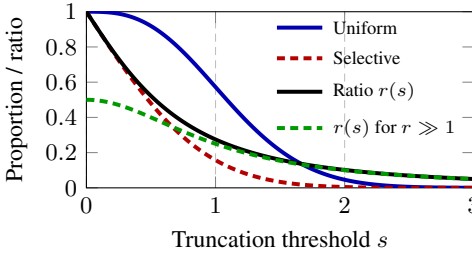 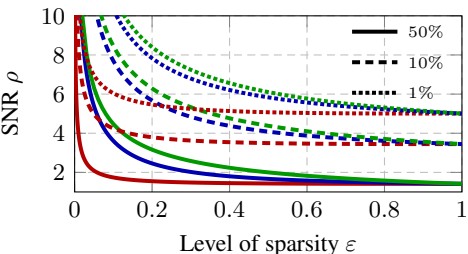

Figure 5: **(Left)** Proportion of nonzero entries with uniform versus selective sparsification $f_1$ and their ratio $r(s)$, as a function of the truncation threshold $s$. **(Right)** Comparison of $1\%, 10\%$ classification error and phase transition (i.e., $50\%$ error) curves between subsampling (**green**), uniform (**blue**) and selective sparsification $f_1$ (**red**), as a function of sparsity level $\varepsilon$ and SNR $\rho$, for $c = 2$.

$r(s) \sim (1 + s^2)^{-1}/2$ for $s \gg 1$,[5] meaning that the gain in storage size and computational time is more significant as the matrix becomes sparser. This is depicted in Figure 5-(left).

Fixing $\alpha$ in Corollary 2 to achieve a given clustering performance level (via Proposition 1), one may then retrieve "equi-performance" curves in the $(\varepsilon, \rho)$-plane, for uniform sparsification, selective sparsification, and subsampling. This is displayed in Figure 5-(right), showing that a dramatic performance gain is achieved by the proposed selective sparsification $f_1$. Besides, here for $c = 2$, as much as $80\%$ sparsity could be obtained with selective sparsification at constant SNR $\rho$, with virtually no performance loss (red curves are almost flat on $\varepsilon \in [0.2, 1]$). This fails to hold for uniform sparsification (Zarrouk et al. (2020) obtain such a result only when $c \lesssim 0.1$) or subsampling.

### 4.3 OPTIMALLY QUANTIZED AND BINARIZED MATRICES

From Figure 4, we see that the classification errors of the quantized $f_2(M; s; t)$ and binarized $f_3(s; t)$ do *not* increase monotonically with the truncation threshold $s$. It can be shown (and also visually confirmed in Figure 4) that, for a given $M \geq 2$, the ratio $\nu/a_1^2$ of both $f_2$ and $f_3$ is convex in $s$ and has a unique minimum. This leads to the following optimal design result for $f_2$ and $f_3$, respectively, the proof of which is straightforward.

**Proposition 2** (Optimal design of quantized and binarized functions). *Under the assumptions and notations of Proposition 1, the classification error rate is minimized at $s = s_{\mathrm{opt}}$ with*

*1. $s_{\mathrm{opt}}$ the unique solution to $a_1(s_{\mathrm{opt}})\nu'(s_{\mathrm{opt}}) = 2a_1'(s_{\mathrm{opt}})\nu(s_{\mathrm{opt}})$ for quantized $f_2$, with $a_1'(s)$ and $\nu'(s)$ the corresponding derivatives with respect to $s$ in Figure 1-(right); and*

*2. $s_{\mathrm{opt}} = \exp(-s_{\mathrm{opt}}^2)/(2\sqrt{\pi}\,\mathrm{erfc}(s_{\mathrm{opt}})) \approx 0.43$ for binary $f_3$, with level of sparsity $\varepsilon \approx 0.54$.*

Therefore, the optimal threshold $s_{\mathrm{opt}}$ for quantized $f_2$ or binary $f_3$ under (1) is *problem-independent*, as it depends neither on $\rho$ nor on $c$. In particular, note that **(i)** the binary $f_3(s_{\mathrm{opt}}; \cdot)$ is *consistently* better than $f(t) = \mathrm{sign}(t)$ for which $\nu/a_1^2 = \pi/2 \approx 1.57 > 1.24$; and **(ii)** the performance of quantized $f_2$ can be *worse*, though very slightly, than that of binary $f_3$ for small $s$, but significantly better for not-too-small $s$. These are visually confirmed in the left and middle displays of Figure 4.

As already observed in Figure 4-(right), a significant gain in storage size can be achieved by using $f_2$ or $f_3$, versus the performance-optimal but dense linear function, with virtually no performance loss. Figure 6 compares the performance of the optimally designed $f_2$ and $f_3$, to sparse $f_1$ that has *approximately* the same storage size.[6] A significant drop in classification error is observed by using quantization $f_2$ or binarization $f_3$ rather than sparsification $f_1$. Also, the performances of $f_2$ and $f_3$ are extremely close to the theoretical optimal (met by $f(t) = t$). This is further confirmed by Figure 6-(right) where, for the optimal $f_2$, the ratio $\nu/a_1^2$ gets close to 1, for all $M \geq 5$.

Figure 7 next evaluates the clustering performance, the proportion of nonzero entries in $\mathbf{K}$, and the computational time of the top eigenvector, for sparse $f_1$ and binary $f_3$, versus linear $f(t) = t$,

---

[5]We use here the asymptotic expansion $\mathrm{erfc}(s) = \frac{e^{-s^2}}{s\sqrt{\pi}}\left[1 + \sum_{k=1}^{\infty}(-1)^k \cdot \frac{1 \cdot 3 \cdots (2k-1)}{(2s^2)^k}\right]$.

[6]We set the truncation threshold $s$ of $f_1$ such that $\mathrm{erfc}(s) = 3/64$, so that the storage size of the sparse $f_1$ (64 bits per nonzero entry) is the same as the quantized $f_2$ with $M = 3$ (with 3 bits per nonzero entry), which is *three times* that of the binary $f_3$.

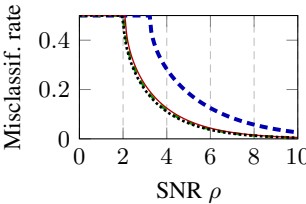 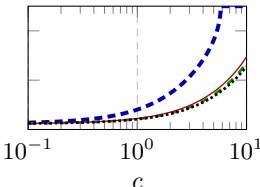 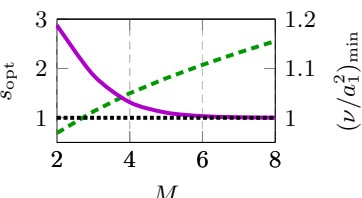

Figure 6: Performance of $f_1$ (**blue**), $f_2$ with $M = 3$ (**green**) and $f_3$ (**red**) of the same storage size, versus SNR for $c = 4$ (**left**) and versus $c$ for SNR $\rho = 4$ (**middle**). **(Right)** Optimal threshold $s_{\mathrm{opt}}$ (**green**) and $(\nu/a_1^2)_{\min}$ (**purple**) of $f_2$ versus $M$. Curves for linear $f(t) = t$ are displayed in **black**.

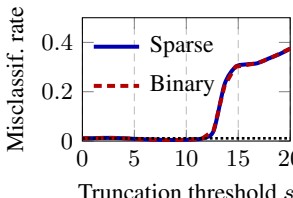 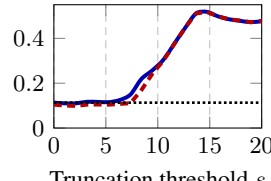 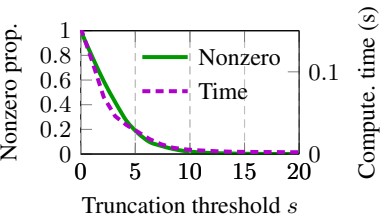

Figure 7: Clustering performance of sparse $f_1$ and binary $f_3$ (**left** and **middle**), proportion of nonzero entries and computational time of the top eigenvector for $f_3$ (**right**), as a function of the truncation threshold $s$ on the MNIST dataset: digits $(0, 1)$ (**left**) and $(5, 6)$ (**middle** and **right**) with $n = 2\,048$ and performance of the linear function in **black**. Results averaged over 100 runs.

as a function of the truncation threshold $s$, on the popular MNIST datasets (LeCun et al., 1998). Depending on (the SNR $\rho$ of) the task, up to $90\%$ of the entries can be discarded almost "for free". Moreover, the curves of the binary $f_3$ appear strikingly close to those of the sparse $f_1$, showing the additional advantage of using the former to further reduce the storage size of **K**. More empirical results on various datasets are provided in Appendix B to confirm our observations in Figure 7.

## 5 CONCLUDING REMARKS

We have evaluated performance-complexity trade-offs when sparsifying, quantizing, and binarizing a linear kernel matrix via a thresholding operator. Our main technical result characterizes the change in the eigenspectrum under these operations; and we have shown that, under an information-plus-noise model, sparsification and quantization, when carefully employed, maintain the informative eigenstructure and incur almost negligible performance loss in spectral clustering. Empirical results on real data demonstrate that these conclusions hold far beyond the present statistical model.

The proposed analysis can be extended in many ways, for instance by considering a multi-cluster and more involved model than (1) as in (Liao & Couillet, 2019) (i.e., "generic" $K$-class Gaussian mixture $\mathcal{N}(\boldsymbol{\mu}_a, \mathbf{C}_a)$ for $a \in \{1, \dots, K\}$, which may help better interpret the empirical observations in Figure 7 and Appendix B), by focusing on more general kernels beyond the current inner-product type in (2), or by deriving non-asymptotic guarantees as in (Vankadara & Ghoshdastidar, 2020).

Our results open the door to theoretical investigation of a broad range of cost-efficient linear algebra methods in machine learning, including subsampling techniques (Mensch et al., 2017; Roosta-Khorasani & Mahoney, 2019), distributed optimization (Wang et al., 2018), randomized linear algebra algorithms (Mahoney, 2011; Drineas & Mahoney, 2016), and quantization for improved training and/or inference (Dong et al., 2019; Shen et al., 2020). Also, given recent interest in viewing neural networks from the perspective of RMT (Li & Nguyen, 2018; Seddik et al., 2018; Jacot et al., 2019; Liu & Dobriban, 2019; Martin & Mahoney, 2019; 2020), our results open the door to understanding and improving performance-complexity trade-offs far beyond kernel methods (Rahimi & Recht, 2008; Jacot et al., 2018; Liu et al., 2020), e.g., to sparse, quantized, or even binary neural networks (Hubara et al., 2016; Lin et al., 2017).

ACKNOWLEDGMENTS

We would like to acknowledge DARPA, IARPA (contract W911NF20C0035), NSF, and ONR via its BRC on RandNLA for providing partial support of this work. Our conclusions do not necessarily reflect the position or the policy of our sponsors, and no official endorsement should be inferred. Couillet's work is partially supported by MIAI at University Grenoble-Alpes (ANR-19-P3IA-0003) and the HUAWEI LarDist project.

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

# A PROOFS AND RELATED DISCUSSIONS

Under the mixture model (1), the data matrix $\mathbf{X} \in \mathbb{R}^{p \times n}$ can be compactly written as

$$\mathbf{X} = \mathbf{Z} + \boldsymbol{\mu}\mathbf{v}^\mathsf{T}, \tag{13}$$

for $\mathbf{Z} \in \mathbb{R}^{p \times n}$ having i.i.d. zero-mean, unit-variance, $\kappa$-kurtosis, sub-exponential entries and $\mathbf{v} \in \{\pm 1\}^n$ so that $\|\mathbf{v}\| = \sqrt{n}$. Recall also the following notations:

$$\mathbf{K} = \left\{ \delta_{i \neq j} f(\mathbf{x}_i^\mathsf{T} \mathbf{x}_j / \sqrt{p}) / \sqrt{p} \right\}_{i,j=1}^n, \quad \mathbf{Q}(z) \equiv (\mathbf{K} - z\mathbf{I}_n)^{-1}. \tag{14}$$

## A.1 PROOF OF THEOREM 1

The proof of Theorem 1 comes in the following two steps:

1. show that the random quantities $\frac{1}{n} \operatorname{tr} \mathbf{A}_n \mathbf{Q}(z)$ and $\mathbf{a}_n^\mathsf{T} \mathbf{Q}(z)\mathbf{b}_n$ of interest concentrate around their expectations in the sense that

$$\frac{1}{n} \operatorname{tr} \mathbf{A}_n (\mathbf{Q}(z) - \mathbb{E}[\mathbf{Q}(z)]) \to 0, \quad \mathbf{a}_n^\mathsf{T} (\mathbf{Q}(z) - \mathbb{E}[\mathbf{Q}(z)])\mathbf{b}_n \to 0, \tag{15}$$

   almost surely as $n, p \to \infty$; and

2. show that the sought-for *deterministic equivalent* $\bar{\mathbf{Q}}(z)$ given in Theorem 1 is an *asymptotic* approximation for the expectation of the resolvent $\mathbf{Q}(z)$ defined in (6) in the sense that

$$\|\mathbb{E}[\mathbf{Q}] - \bar{\mathbf{Q}}\| \to 0, \tag{16}$$

   as $n, p \to \infty$.

The concentration of trace forms in the first item has been established in (Cheng & Singer, 2013; Do & Vu, 2013), and the bilinear forms follow similarly. Here we focus on the second item to show that $\|\mathbb{E}[\mathbf{Q}] - \bar{\mathbf{Q}}\| \to 0$ in the large $n, p$ limit.

In the sequel, we use $o(1)$ and $o_{\|\cdot\|}(1)$ for scalars or matrices of (almost surely if being random) vanishing absolute values or operator norms as $n, p \to \infty$.

To establish $\|\mathbb{E}[\mathbf{Q}] - \bar{\mathbf{Q}}\| \to 0$, we need to show subsequently that:

1. under (13), the random matrix $\mathbf{K}$ defined in (2) admits a spiked-model approximation, that is

$$\mathbf{K} = \tilde{\mathbf{K}}_0 + \mathbf{U}\boldsymbol{\Lambda}\mathbf{U}^\mathsf{T} + o_{\|\cdot\|}(1), \tag{17}$$

for some full rank random (noise) matrix $\tilde{\mathbf{K}}_0$ and low rank (information) matrix $\mathbf{U}\boldsymbol{\Lambda}\mathbf{U}^\mathsf{T}$ to be specified; and

2. the matrix inverse $(\tilde{\mathbf{K}}_0 - \mathbf{U}\boldsymbol{\Lambda}\mathbf{U}^\mathsf{T} - z\mathbf{I}_n)^{-1}$ can be decomposed with the Woodbury identity, so that

$$\mathbf{Q} = (\tilde{\mathbf{K}}_0 - \mathbf{U}\boldsymbol{\Lambda}\mathbf{U}^\mathsf{T} - z\mathbf{I}_n)^{-1} + o_{\|\cdot\|}(1) = \tilde{\mathbf{Q}}_0 - \tilde{\mathbf{Q}}_0 \mathbf{U}(\boldsymbol{\Lambda}^{-1} + \mathbf{U}^\mathsf{T}\tilde{\mathbf{Q}}_0\mathbf{U})^{-1}\mathbf{U}^\mathsf{T}\tilde{\mathbf{Q}}_0 + o_{\|\cdot\|}(1), \tag{18}$$

with $\tilde{\mathbf{Q}}_0(z) \equiv (\tilde{\mathbf{K}}_0 - z\mathbf{I}_n)^{-1}$; and

3. the expectation of the right-hand side of (18) is close to $\bar{\mathbf{Q}}$ in the large $n, p$ limit, allowing us to conclude the proof of Theorem 1.

To establish (17), we denote the "noise-only" null model with $\|\boldsymbol{\mu}\| = 0$ by writing $\mathbf{K} = \mathbf{K}_0$ such that

$$[\mathbf{K}_0]_{ij} = \delta_{i \neq j} f(\mathbf{z}_i^\mathsf{T} \mathbf{z}_j / \sqrt{p}) / \sqrt{p}. \tag{19}$$

With a combinatorial argument, it has been shown in (Fan & Montanari, 2019) that

$$\left\| \mathbf{K}_0 - \tilde{\mathbf{K}}_0 - \frac{a_2}{\sqrt{2}} \frac{1}{p} (\boldsymbol{\psi}\mathbf{1}_n^\mathsf{T} + \mathbf{1}_n\boldsymbol{\psi}^\mathsf{T}) \right\| \to 0, \tag{20}$$

almost surely as $n, p \to \infty$, for $\tilde{\mathbf{K}}_0$ such that $(\tilde{\mathbf{K}}_0 - z\mathbf{I}_n)^{-1} \equiv \tilde{\mathbf{Q}}_0(z) \leftrightarrow m(z)\mathbf{I}_n$ and the random vector $\boldsymbol{\psi} \in \mathbb{R}^n$ with its $i$-th entries given by

$$[\boldsymbol{\psi}]_i = \frac{1}{\sqrt{p}} \left( \|\mathbf{z}_i\|^2 - \mathbb{E}[\|\mathbf{z}_i\|^2] \right) = \frac{1}{\sqrt{p}} (\|\mathbf{z}_i\|^2 - p).$$

Consider now the informative-plus-noise model $\mathbf{K}$ for $\mathbf{X} = \mathbf{Z} + \boldsymbol{\mu}\mathbf{v}^\mathsf{T}$ as in (13) with $[\mathbf{v}]_i = \pm 1$ and $\|\mathbf{v}\| = \sqrt{n}$. It follows from (Liao & Couillet, 2019) that

$$\left\| \mathbf{K} - \mathbf{K}_0 - \frac{a_1}{p} \begin{bmatrix} \mathbf{v} & \mathbf{Z}^\mathsf{T}\boldsymbol{\mu} \end{bmatrix} \begin{bmatrix} \|\boldsymbol{\mu}\|^2 & 1 \\ 1 & 0 \end{bmatrix} \begin{bmatrix} \mathbf{v}^\mathsf{T} \\ \boldsymbol{\mu}^\mathsf{T}\mathbf{Z} \end{bmatrix} \right\| \to 0, \tag{21}$$

almost surely as $n, p \to \infty$.

Combining (20) with (21), we obtain $\|\mathbf{K} - \tilde{\mathbf{K}}_0 - \mathbf{U}\boldsymbol{\Lambda}\mathbf{U}^\mathsf{T}\| \to 0$ almost surely as $n, p \to \infty$, with

$$\mathbf{U} = \frac{1}{\sqrt{p}}[\mathbf{1}_n, \ \mathbf{v}, \ \boldsymbol{\psi}, \ \mathbf{Z}^\mathsf{T}\boldsymbol{\mu}] \in \mathbb{R}^{n \times 4}, \quad \boldsymbol{\Lambda} = \begin{bmatrix} 0 & 0 & \frac{a_2}{\sqrt{2}} & 0 \\ 0 & a_1\|\boldsymbol{\mu}\|^2 & 0 & a_1 \\ \frac{a_2}{\sqrt{2}} & 0 & 0 & 0 \\ 0 & a_1 & 0 & 0 \end{bmatrix} \tag{22}$$

and $(\tilde{\mathbf{K}}_0 - z\mathbf{I}_n)^{-1} \equiv \tilde{\mathbf{Q}}_0(z) \leftrightarrow m(z)\mathbf{I}_n$. By the Woodbury identity, we write

$$\mathbf{Q} = (\mathbf{K} - z\mathbf{I}_n)^{-1} = (\tilde{\mathbf{K}}_0 + \mathbf{U}\boldsymbol{\Lambda}\mathbf{U}^\mathsf{T} - z\mathbf{I}_n)^{-1} + o_{\|\cdot\|}(1)$$
$$= \tilde{\mathbf{Q}}_0 - \tilde{\mathbf{Q}}_0\mathbf{U}(\boldsymbol{\Lambda}^{-1} + \mathbf{U}^\mathsf{T}\tilde{\mathbf{Q}}_0\mathbf{U})^{-1}\mathbf{U}^\mathsf{T}\tilde{\mathbf{Q}}_0 + o_{\|\cdot\|}(1) \tag{23}$$

with

$$\boldsymbol{\Lambda}^{-1} + \mathbf{U}^\mathsf{T}\tilde{\mathbf{Q}}_0\mathbf{U} = \begin{bmatrix} \frac{m(z)}{c} & \frac{m(z)}{c}\frac{\mathbf{v}^\mathsf{T}\mathbf{1}_n}{n} & 0 & 0 \\ \frac{m(z)}{c}\frac{\mathbf{v}^\mathsf{T}\mathbf{1}_n}{n} & \frac{m(z)}{c} & 0 & 0 \\ 0 & 0 & (\kappa - 1)\frac{m(z)}{c} & 0 \\ 0 & 0 & 0 & \boldsymbol{\mu}^\mathsf{T}(\frac{1}{p}\mathbb{E}[\mathbf{Z}\tilde{\mathbf{Q}}_0\mathbf{Z}^\mathsf{T}])\boldsymbol{\mu} \end{bmatrix} + o_{\|\cdot\|}(1)$$

where we use the fact that

$$\mathbb{E}[\boldsymbol{\psi}] = \mathbf{0}, \quad \mathbb{E}[\boldsymbol{\psi}\boldsymbol{\psi}^\mathsf{T}] = (\kappa - 1)\mathbf{I}_n.$$

We need to evaluate the expectation $\frac{1}{p}\mathbb{E}[\mathbf{Z}(\tilde{\mathbf{K}}_0 - z\mathbf{I}_n)^{-1}\mathbf{Z}^\mathsf{T}]$. This is given in the following lemma.

**Lemma 1.** *Under the assumptions and notations of Theorem 1, we have*

$$\frac{1}{p}\mathbb{E}[\mathbf{Z}(\tilde{\mathbf{K}}_0 - z\mathbf{I}_n)^{-1}\mathbf{Z}^\mathsf{T}] = \frac{m(z)}{c + a_1 m(z)}\mathbf{I}_p + o_{\|\cdot\|}(1). \tag{24}$$

*Proof of Lemma 1.* For $\tilde{\mathbf{Q}}_0 = (\tilde{\mathbf{K}}_0 - z\mathbf{I}_n)^{-1}$, we aim to approximate the expectation $\mathbb{E}[\mathbf{Z}\tilde{\mathbf{Q}}_0\mathbf{Z}^\mathsf{T}]/p$. Consider first the case where the entries of $\mathbf{Z}$ are i.i.d. Gaussian, we can write the $(i, i')$ entry of $\mathbb{E}[\mathbf{Z}\tilde{\mathbf{Q}}_0\mathbf{Z}^\mathsf{T}]$ with Stein's lemma (i.e., $\mathbb{E}[xf(x)] = \mathbb{E}[f'(x)]$ for $x \sim \mathcal{N}(0, 1)$) as

$$\mathbb{E}[\mathbf{Z}\tilde{\mathbf{Q}}_0\mathbf{Z}^\mathsf{T}]_{ii'} = \sum_{j=1}^n \mathbb{E}[\mathbf{Z}_{ij}[\tilde{\mathbf{Q}}_0\mathbf{Z}^\mathsf{T}]_{ji'}] = \sum_{j=1}^n \mathbb{E}\frac{\partial[\tilde{\mathbf{Q}}_0\mathbf{Z}^\mathsf{T}]_{ji'}}{\partial\mathbf{Z}_{ij}}$$
$$= \sum_{j=1}^n \mathbb{E}\left[[\tilde{\mathbf{Q}}_0]_{jj}\delta_{ii'} + \sum_{k=1}^n \frac{\partial[\tilde{\mathbf{Q}}_0]_{jk}}{\partial\mathbf{Z}_{ij}}\mathbf{Z}_{ki'}^\mathsf{T}\right].$$

We first focus on the term $\frac{\partial[\tilde{\mathbf{Q}}_0]_{jk}}{\partial\mathbf{Z}_{ij}}$ by writing

$$\frac{\partial[\tilde{\mathbf{Q}}_0]_{jk}}{\partial\mathbf{Z}_{ij}} = -\left[\tilde{\mathbf{Q}}_0\frac{\partial\mathbf{K}_0}{\partial\mathbf{Z}_{ij}}\tilde{\mathbf{Q}}_0\right]_{jk} = \sum_{l,m=1}^n -[\tilde{\mathbf{Q}}_0]_{jl}\frac{\partial[\mathbf{K}_0]_{lm}}{\partial\mathbf{Z}_{ij}}[\tilde{\mathbf{Q}}_0]_{mk}$$

where we recall $[\mathbf{K}_0]_{ij} = \delta_{i\neq j}f(\mathbf{Z}^\mathsf{T}\mathbf{Z}/\sqrt{p})_{ij}/\sqrt{p}$ so that for $l \neq m$ we have

$$\frac{\partial[\mathbf{K}_0]_{lm}}{\partial\mathbf{Z}_{ij}} = \frac{1}{p}f'(\mathbf{Z}^\mathsf{T}\mathbf{Z}/\sqrt{p})_{lm}\frac{\partial[\mathbf{Z}^\mathsf{T}\mathbf{Z}]_{lm}}{\partial\mathbf{Z}_{ij}} = \frac{1}{p}f'(\mathbf{Z}^\mathsf{T}\mathbf{Z}/\sqrt{p})_{lm}(\delta_{jl}\mathbf{Z}_{im} + \mathbf{Z}_{li}^\mathsf{T}\delta_{jm})$$

and $\frac{\partial[\mathbf{K}_0]_{lm}}{\partial\mathbf{Z}_{ij}} = 0$ for $l = m$. We get

$$\sum_{j,k}\frac{\partial[\tilde{\mathbf{Q}}_0]_{jk}}{\partial\mathbf{Z}_{ij}}\mathbf{Z}_{ki'}^\mathsf{T} = -\frac{1}{p}\sum_{j,k,m}[\tilde{\mathbf{Q}}_0]_{jj}f'(\mathbf{Z}^\mathsf{T}\mathbf{Z}/\sqrt{p})_{jm}\mathbf{Z}_{im}[\tilde{\mathbf{Q}}_0]_{mk}\mathbf{Z}_{ki'}^\mathsf{T} - \frac{1}{p}\sum_{j,k,l}[\tilde{\mathbf{Q}}_0]_{jl}f'(\mathbf{Z}^\mathsf{T}\mathbf{Z}/\sqrt{p})_{lj}\mathbf{Z}_{li}^\mathsf{T}[\tilde{\mathbf{Q}}_0]_{jk}\mathbf{Z}_{ki'}^\mathsf{T}$$

$$= -\frac{1}{p}[\mathbf{Z}\operatorname{diag}(f'(\mathbf{Z}^\mathsf{T}\mathbf{Z}/\sqrt{p})\tilde{\mathbf{Q}}_0\mathbf{1}_n)\tilde{\mathbf{Q}}_0\mathbf{Z}^\mathsf{T}]_{ii'} - \frac{1}{p}[\mathbf{Z}(\tilde{\mathbf{Q}}_0 \odot f'(\mathbf{Z}^\mathsf{T}\mathbf{Z}/\sqrt{p}))\tilde{\mathbf{Q}}_0\mathbf{Z}^\mathsf{T}]_{ii'}$$

where $f'(\mathbf{Z}^\mathsf{T}\mathbf{Z}/\sqrt{p})$ indeed represents $f'(\mathbf{Z}^\mathsf{T}\mathbf{Z}/\sqrt{p}) - \text{diag}(\cdot)$ in both cases.

For the first term, since $f'(\mathbf{Z}^\mathsf{T}\mathbf{Z}/\sqrt{p}) - \text{diag}(\cdot) = a_1 \mathbf{1}_n \mathbf{1}_n^\mathsf{T} + O_{\|\cdot\|}(\sqrt{p})$, we have

$$\frac{1}{p} f'(\mathbf{Z}^\mathsf{T}\mathbf{Z}/\sqrt{p})\tilde{\mathbf{Q}}_0 \mathbf{1}_n = \frac{a_1}{p} \mathbf{1}_n^\mathsf{T} \tilde{\mathbf{Q}}_0 \mathbf{1}_n \cdot \mathbf{1}_n + O(p^{-1/2}) = \frac{a_1 m(z)}{c} \mathbf{1}_n + O(p^{-1/2}) \quad (25)$$

where $O(p^{-1/2})$ is understood entry-wise. As a result,

$$\frac{1}{p} \mathbf{Z}\, \text{diag}(f'(\mathbf{Z}^\mathsf{T}\mathbf{Z}/\sqrt{p})\tilde{\mathbf{Q}}_0 \mathbf{1}_n)\tilde{\mathbf{Q}}_0 \mathbf{Z}^\mathsf{T} = \frac{a_1 m(z)}{c} \cdot \frac{1}{p} \mathbf{Z}\tilde{\mathbf{Q}}_0 \mathbf{Z}^\mathsf{T} + o_{\|\cdot\|}(1).$$

For the second term, since $f'(\mathbf{Z}^\mathsf{T}\mathbf{Z}/\sqrt{p})$ has $O(1)$ entries and $\|\mathbf{A} \odot \mathbf{B}\| \leq \sqrt{n}\|\mathbf{A}\|_\infty \|\mathbf{B}\|$ for $\mathbf{A}, \mathbf{B} \in \mathbb{R}^{n\times n}$, we deduce that

$$\frac{1}{p}\|\mathbf{Z}(\tilde{\mathbf{Q}}_0 \odot f'(\mathbf{Z}^\mathsf{T}\mathbf{Z}/\sqrt{p}))\tilde{\mathbf{Q}}_0 \mathbf{Z}^\mathsf{T}\| = O(\sqrt{p}).$$

As a consequence, we conclude that

$$\frac{1}{p}\mathbb{E}[\mathbf{Z}\tilde{\mathbf{Q}}_0 \mathbf{Z}^\mathsf{T}] = \frac{1}{p}\,\text{tr}\,\tilde{\mathbf{Q}}_0 \cdot \mathbf{I}_p - \frac{a_1 m(z)}{c} \cdot \frac{1}{p}\mathbb{E}[\mathbf{Z}\tilde{\mathbf{Q}}_0 \mathbf{Z}^\mathsf{T}] + o_{\|\cdot\|}(1)$$

that is

$$\frac{1}{p}\mathbb{E}[\mathbf{Z}\tilde{\mathbf{Q}}_0 \mathbf{Z}^\mathsf{T}] = \frac{m(z)}{c + a_1 m(z)}\mathbf{I}_p + o_{\|\cdot\|}(1)$$

where we recall that $\text{tr}\,\tilde{\mathbf{Q}}_0/p = m(z)/c$ and thus the conclusion of Lemma 1 for the Gaussian case. The interpolation trick (Lytova & Pastur, 2009, Corollaray 3.1) can then be applied to extend the result beyond Gaussian distribution. This concludes the proof of Lemma 1. $\qquad\square$

Denote $\mathbf{A} = (\mathbf{\Lambda}^{-1} + \mathbf{U}^\mathsf{T}\tilde{\mathbf{Q}}_0 \mathbf{U})^{-1}$, it follows from Lemma 1 that

$$\mathbb{E}[\mathbf{U}\mathbf{A}\mathbf{U}^\mathsf{T}] = \frac{1}{p}(\mathbf{A}_{11}\mathbf{1}_n \mathbf{1}_n^\mathsf{T} + \mathbf{A}_{12}\mathbf{1}_n \mathbf{v}^\mathsf{T} + \mathbf{A}_{21}\mathbf{v}\mathbf{1}_n^\mathsf{T} + \mathbf{A}_{22}\mathbf{v}\mathbf{v}^\mathsf{T} + \mathbf{A}_{33}(\kappa - 1)\mathbf{I}_n + \mathbf{A}_{44}\|\boldsymbol{\mu}\|^2 \mathbf{I}_n)$$

$$= \frac{1}{p}(\mathbf{A}_{11}\mathbf{1}_n \mathbf{1}_n^\mathsf{T} + \mathbf{A}_{12}\mathbf{1}_n \mathbf{v}^\mathsf{T} + \mathbf{A}_{21}\mathbf{v}\mathbf{1}_n^\mathsf{T} + \mathbf{A}_{22}\mathbf{v}\mathbf{v}^\mathsf{T}) + o_{\|\cdot\|}(1)$$

since $\|\boldsymbol{\mu}\| = O(1)$ and $\|\mathbf{v}\| = O(\sqrt{n})$. We thus deduce from (23) that

$$\mathbf{Q}(z) \leftrightarrow \bar{\mathbf{Q}}(z) = m(z)\mathbf{I}_n - cm^2(z)\mathbf{V}\begin{bmatrix} \mathbf{A}_{11} & \mathbf{A}_{12} \\ \mathbf{A}_{21} & \mathbf{A}_{22} \end{bmatrix}\mathbf{V}^\mathsf{T}$$

with $\sqrt{n}\mathbf{V} = [\mathbf{v}, \ \mathbf{1}_n]$. Rearranging the expression we conclude the proof of Theorem 1. $\qquad\square$

## A.2 PROOF OF COROLLARY 1 AND RELATED DISCUSSIONS

Consider the noise-only model by taking $\boldsymbol{\mu} = \mathbf{0}$ in Theorem 1. Then, we have $\mathbf{K} = \mathbf{K}_0$ and $\Theta(z) = 0$, so that

$$\bar{\mathbf{Q}}(z) = m(z) + \Omega(z) \cdot \frac{1}{n}\mathbf{1}_n \mathbf{1}_n^\mathsf{T}, \quad \Omega(z) = \frac{a_2^2(\kappa - 1)m^3(z)}{2c^2 - a_2^2(\kappa - 1)m^2(z)} \quad (26)$$

where we recall $m(z)$ is the solution to

$$m(z) = -\left(z + \frac{a_1^2 m(z)}{c + a_1 m(z)} + \frac{\nu - a_1^2}{c}m(z)\right)^{-1}. \quad (27)$$

Since the resolvent $\mathbf{Q}(z)$ is undefined for $z \in \mathbb{R}$ within the eigensupport of $\mathbf{K}$ that consists of (i) the main bulk characterized by the Stieltjes transform $m(z)$ defined in (27) and (ii) the possible spikes, we need to find the poles of $\bar{\mathbf{Q}}(z)$ but not those of $m(z)$ to determine the asymptotic locations of the spikes that are *away from* the main bulk. Direct calculations show that the Stieltjes transforms of the possible *non-informative* spikes satisfy

$$m_\pm = \pm\sqrt{\frac{2}{\kappa - 1}}\frac{c}{a_2} \quad (28)$$

that are in fact the poles of $\Omega(z)$, for $a_2 \neq 0$ and $\kappa \neq 1$. For $\kappa = 1$ or $a_2 = 0$, $\Omega(z)$ has no (additional) poles, so that there is (almost surely) no spike outside the limiting spectrum.

It is however not guaranteed that $z \in \mathbb{R}$ corresponding to (28) isolates from the main bulk. To this end, we introduce the following characterization of the limiting spectral measure in (27), the proof of which follows from previous work.

**Corollary 3** (Limiting spectrum). *Under the notations and conditions of Theorem 1, with proba-bility one, the empirical spectral measure $\omega_n = \frac{1}{n} \sum_{i=1}^{n} \delta_{\lambda_i(\mathbf{K}_0)}$ of the noise-only model $\mathbf{K}_0$ (and therefore that of $\mathbf{K}$ as a low rank additive perturbation of $\mathbf{K}_0$ via (21)) converges weakly to a prob-ability measure $\omega$ of compact support as $n, p \to \infty$, with $\omega$ uniquely defined through its Stieltjes transform $m(z)$ solution to (27). Moreover,*

1. *if we let $\operatorname{supp}(\omega)$ be the support of $\omega$, then*

$$\operatorname{supp}(\omega) \cup \{0\} = \mathbb{R} \setminus \{x(m) \mid m \in \mathbb{R} \setminus \{\{-c/a_1\} \cup \{0\}\} \text{ and } x'(m) > 0\} \quad (29)$$

   *for $x(m)$ the functional inverse of (27) explicitly given by*

$$x(m) = -\frac{1}{m} - \frac{a_1^2 m}{c + a_1 m} - \frac{\nu - a_1^2}{c} m, \quad (30)$$

2. *the measure $\omega$ has a density and its support may have up to four edges, with the associated Stieltjes transforms given by the roots of $x'(m) = 0$, i.e.,*

$$x'(m) = \frac{1}{m^2} - \frac{a_1^2 c}{(c + a_1 m)^2} - \frac{\nu - a_1^2}{c} = 0. \quad (31)$$

The limiting spectral measure $\omega$ of the null model $\mathbf{K}_0$ was first derived in (Cheng & Singer, 2013) for Gaussian distribution and then extended to sub-exponential distribution in (Do & Vu, 2013). The fact that a finite rank perturbation does not affect the limiting spectrum follows from (Silverstein & Bai, 1995, Lemma 2.6).

The characterization in (29) above follows the same idea as in (Silverstein & Choi, 1995, Theo-rem 1.1), which arises from the crucial fact that the Stieltjes transform $m(x) = \int (t - x)^{-1} \omega(dt)$ of a measure $\omega$ is an increasing function on its domain of definition and so must be its functional inverse $x(m)$ given explicitly in (30). In plain words, Corollary 3 tells us that (**i**) depending on the number of real solutions to (31), the support of $\omega$ may contain two *disjoint* regions with four edges, and (**ii**) $x \in \mathbb{R}$ is outside the support of $\omega$ *if and only if* its associated Stieltjes transform $m$ satisfies $x'(m) > 0$, i.e., belonging to the increasing region of the functional inverse $x(m)$ in (30). This is depicted in Figure 8, where for the same function $f(t) = \max(t, 0) - 1/\sqrt{2\pi}$ with $a_1 = 1/2$, $a_2 = 1/(2\sqrt{\pi})$ and $\nu = (\pi - 1)/(2\pi)$, we observe in the top display a single region of $\omega$ for $c = 2$ and in the bottom display two disjoint regions (with thus four edges) for $c = 1/10$. The correspond-ing (empirical) eigenvalue histograms and limiting laws are given in Figure 9. Note, in particular, that the local extrema of the functional inverse $x(m)$ in Figure 8 characterize the (possibly up to four) edges of the support of $\omega$ in Figure 9.

According to the discussion above, it remains to check the sign of $x'(m)$ for $m = \pm \sqrt{\frac{2}{\kappa-1}} \frac{c}{a_2}$ to see if they correspond to isolated eigenvalues away from the support of $\omega$. This, after some algebraic manipulations, concludes the proof of Corollary 1. $\qquad \square$

**Discussions.** The limiting spectral measure in Corollary 3 is indeed a "mix" between the popular Marčenko-Pastur and the Wigner's semicircle law.

**Remark 2** (From Marčenko-Pastur to semicircle law). *As already pointed out in (Fan & Montanari, 2019), here the limiting spectral measure $\omega$ is the so-called* free additive convolution *(Voiculescu, 1986) of the semicircle and Marčenko-Pastur laws, weighted respectively by $a_1$ and $\sqrt{\nu - a_1^2}$, i.e.,*

$$\omega = a_1(\omega_{MP,c^{-1}} - 1) \boxplus \sqrt{(\nu - a_1^2)/c} \cdot \omega_{SC} \quad (32)$$

*where we denote $a_1(\omega_{MP,c^{-1}} - 1)$ the law of $a_1(x - 1)$ for $x \sim \omega_{MP,c^{-1}}$ and $\sqrt{(\nu - a_1^2)/c} \cdot \omega_{SC}$ the law of $\sqrt{(\nu - a_1^2)/c} \cdot x$ for $x \sim \omega_{SC}$. Figure 10 compares the eigenvalue distributions of $\mathbf{K}_0$*

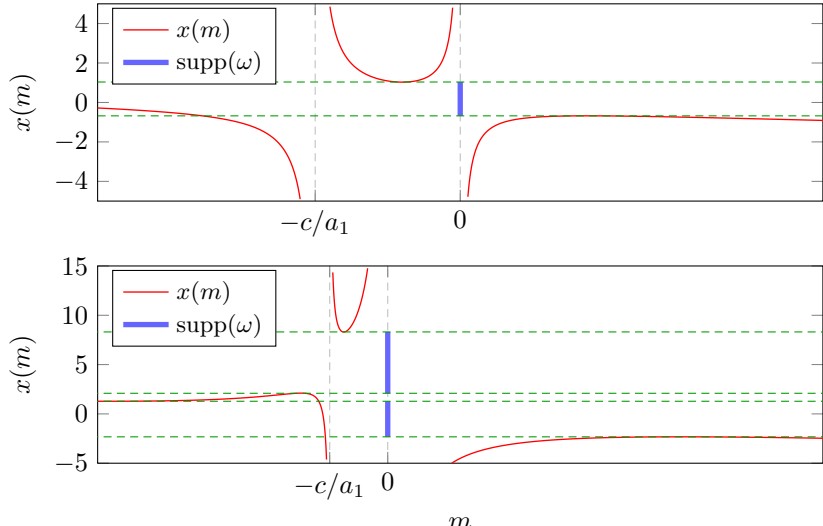

Figure 8: Functional inverse $x(m)$ for $m \in \mathbb{R} \setminus \{\{-c/a_1\} \cup \{0\}\}$, with $f(t) = \max(t,0) - 1/\sqrt{2\pi}$, for $c = 2$ (**above**, with two edges) and $c = 1/10$ (**bottom**, with four edges). The support of $\omega$ can be read on the vertical axes and the values of $x$ such that $x'(m) = 0$ are marked in **green**.

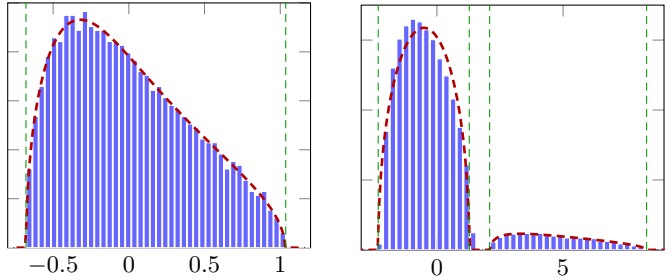

Figure 9: Eigenvalues of $\mathbf{K}$ with $\boldsymbol{\mu} = \mathbf{0}$ (**blue**) versus the limiting laws in Theorem 1 and Corollary 3 (**red**) for $p = 3\,200$, $n = 1\,600$ (**left**) and $p = 400$, $n = 4\,000$ (**right**), with $f(t) = \max(t,0) - 1/\sqrt{2\pi}$ and Gaussian data. The values of $x$ such that $x'(m) = 0$ in Figure 8 are marked in **green**.

*for $f(t) = a_1 t + a_2(t^2 - 1)/\sqrt{2}$ (so that $\nu - a_1^2 = a_2^2$) with different pairs of $(a_1, a_2)$. We observe a transition from the Marčenko-Pastur law (in the left display, with $a_1 \neq 0$ and $a_2 = 0$) to the semicircle law (in the right display, with $a_1 = 0$ and $a_2 \neq 0$).*

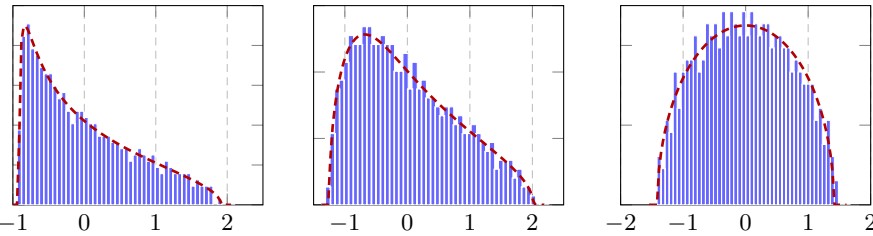

Figure 10: Eigenvalues of $\mathbf{K}$ with $\boldsymbol{\mu} = \mathbf{0}$ (**blue**) versus the limiting laws in Theorem 1 and Corollary 3 (**red**) for Gaussian data, $p = 1\,024$, $n = 512$ and $f(t) = a_1 t + a_2(t^2 - 1)/\sqrt{2}$ with $a_1 = 1, a_2 = 0$ (**left**), $a_1 = 1, a_2 = 1/2$ (**middle**), and $a_1 = 0, a_2 = 1$ (**right**).

Remark 2 tells us that, depending on the ratio $\nu/a_1^2$, the eigenspectrum of $\mathbf{K}$ exhibits a transition from the Marčenko-Pastur to semicircle-like shape. Note from Figure 1-(right) that, for the sparse $f_1$, the ratio $\nu/a_1^2$ is an increasing function of the truncation threshold $s$ and therefore, as the matrix $\mathbf{K}$ become sparser, it eigenspectrum changes from a Marčenko-Pastur-type (at $s = 0$) to be more semicircle-like. This is depicted in Figure 11 and similar conclusions hold for quantized $f_2$ and binary $f_3$ in the $s \geq s_{\text{opt}}$ regime.

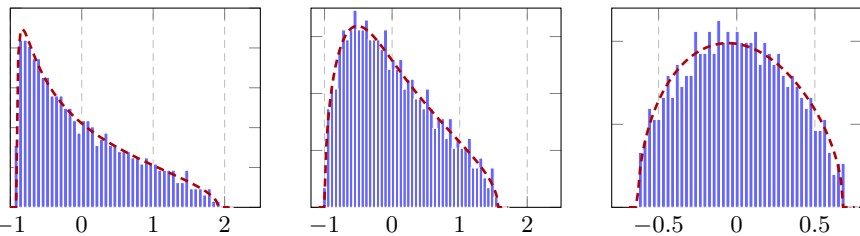

Figure 11: Eigenvalues of $\mathbf{K}$ with $\boldsymbol{\mu} = \mathbf{0}$ (**blue**) versus the limiting laws in Theorem 1 and Corollary 3 (**red**) for Gaussian data, $p = 1\,024$, $n = 512$ and $f(t) = t \cdot 1_{|t|>\sqrt{2}s}$ with $s = 0.1$ (**left**), $s = .75$ (**middle**), and $s = 1.5$ (**right**).

As discussed after Theorem 1 and in the proof above, while the limiting eigenvalue distribution $\omega$ is *universal* and *independent* of the law of the entries of $\mathbf{Z}$, so long as they are independent, sub-exponential, of zero mean and unit variance, as commonly observed in RMT (Tao et al., 2010), this is no longer the case for the isolated eigenvalues. In particular, according to Corollary 1, the possible non-informative spikes *do depend* on the kurtosis $\kappa$ of the distribution. In Figure 12 we observe a farther (left) spike for Student-t (with $\kappa = 5$ and is thus *not* sub-exponential) than Gaussian distribution (with $\kappa = 3$), while *no* spike can be observed for the symmetric Bernoulli distribution (that takes values $\pm 1$ with probability $1/2$ so that $\kappa = 1$), with the same limiting eigenvalue distribution for $f(t) = \max(t, 0) - 1/\sqrt{2\pi}$.

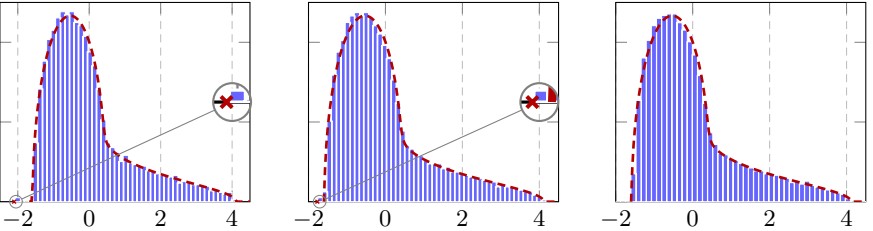

Figure 12: Eigenvalues of $\mathbf{K}$ with $\boldsymbol{\mu} = \mathbf{0}$ (**blue**) versus the limiting laws and spikes in Theorem 1 and Corollary 1 (**red**) for Student-t (with 7 degrees of freedom, **left**), Gaussian (**middle**) and Rademacher distribution (**right**), $p = 512$, $n = 2\,048$, $f(t) = \max(t, 0) - 1/\sqrt{2\pi}$. Emphasis on the *non-informative* spikes at *different* locations: at $-\mathbf{2.10}$ for Student-t and $-\mathbf{1.77}$ for Gaussian.

**Remark 3** (Non-informative spike in-between). *When the support of $\omega$ consists of two disjoint regions (e.g., in the right plot of Figure 9), a non-informative spike may appear between these two regions, with the associated Stieltjes transform $m < -c/a_1$ in the setting of Figure 8-(bottom). This is only possibly when $a_1 \sqrt{\frac{2}{\kappa-1}} > a_2$. An example is provided in Figure 13.*

## A.3   Proof of Corollary 2 and related discussions

Similar to our discussions in Section A.2, we need to find the zeros of $\det \mathbf{\Lambda}(z)$, that are real solutions to $H(x) = 0$ with

$$H(x) = a_1 a_2^2 (\kappa-1) \left( \frac{(\mathbf{v}^\mathsf{T} \mathbf{1}_n)^2}{n^2} \rho - 1 - \rho \right) m^3(x) - a_2^2 c (\kappa-1) m^2(x) + 2 a_1 c^2 (\rho+1) m(x) + 2 c^3 = 0 \tag{33}$$

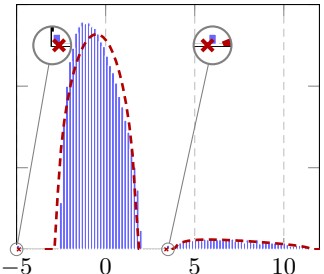

Figure 13: Eigenvalues of $\mathbf{K}$ with $\boldsymbol{\mu} = \mathbf{0}$ (**blue**) versus the limiting laws and spikes in Corollary 3 and 1 (**red**) for $p = 400$, $n = 6\,000$, with $f(t) = \max(t, 0) - 1/\sqrt{2\pi}$ and Gaussian data.

for $m(z)$ the unique solution to (9) and $\rho = \lim_p \|\boldsymbol{\mu}\|^2$. Note that

1. for $a_1 a_2^2 (\kappa - 1)(\frac{(\mathbf{v}^\mathsf{T} \mathbf{1}_n)^2}{n^2} \rho - 1 - \rho) \neq 0$, there can be up to three spikes;

2. with $a_1 = 0$ and $a_2 \neq 0$, we get $m^2(x) = \frac{2c^2}{a_2^2(\kappa-1)}$ and there are at most two spikes: this is equivalent to the case of Corollary 1 with $\rho = 0$; in fact, taking $a_1$ we *discard* the information in the signal $\boldsymbol{\mu}$, as has been pointed out in (Liao & Couillet, 2019);

3. with $a_2 = 0$ and $a_1 \neq 0$ we obtain $m(x) = -\frac{c}{a_1(\rho+1)}$, this is the case of Corollary 2.

For a given isolated eigenvalue-eigenvector pair $(\hat{\lambda}, \hat{\mathbf{v}})$ (assumed to be of multiplicity one), the projection $|\hat{\mathbf{v}}^\mathsf{T} \mathbf{v}|^2$ onto the label vector $\mathbf{v}$ can be evaluated via the Cauchy's integral formula and our Theorem 1. More precisely, consider a positively oriented contour $\Gamma$ that circles around *only* the isolated $\hat{\lambda}$, we write

$$
\frac{1}{n} \mathbf{v}^\mathsf{T} \hat{\mathbf{v}} \hat{\mathbf{v}}^\mathsf{T} \mathbf{v} = -\frac{1}{2\pi\imath} \oint_\Gamma \frac{1}{n} \mathbf{v}^\mathsf{T} (\mathbf{K} - z\mathbf{I}_n)^{-1} \mathbf{v}\, dz
$$

$$
= -\frac{1}{2\pi\imath} \oint_\Gamma \frac{1}{n} \mathbf{v}^\mathsf{T} (m(z)\mathbf{I}_n - \mathbf{V}\boldsymbol{\Lambda}(z)\mathbf{V}^\mathsf{T}) \mathbf{v}\, dz + o(1)
$$

$$
= \frac{1}{n} \mathbf{v}^\mathsf{T} \mathbf{V} \left( \frac{1}{2\pi\imath} \oint_\Gamma \boldsymbol{\Lambda}(z)\, dz \right) \mathbf{V}^\mathsf{T} \mathbf{v} + o(1) = \frac{1}{n} \mathbf{v}^\mathsf{T} \mathbf{V} \left( \mathrm{Res}\boldsymbol{\Lambda}(z) \right) \mathbf{V}^\mathsf{T} \mathbf{v} + o(1)
$$

$$
= \begin{bmatrix} 1 & \frac{\mathbf{v}^\mathsf{T} \mathbf{1}_n}{n} \end{bmatrix} \left( \lim_{z \to \lambda} (z - \lambda) \begin{bmatrix} \Theta(z)m^2(z) & \Theta(z)\Omega(z)\frac{\mathbf{v}^\mathsf{T}\mathbf{1}_n}{n} m(z) \\ \Theta(z)\Omega(z)\frac{\mathbf{v}^\mathsf{T}\mathbf{1}_n}{n} m(z) & \Theta(z)\Omega^2(z)\frac{(\mathbf{v}^\mathsf{T}\mathbf{1}_n)^2}{n^2} - \Omega(z) \end{bmatrix} \right) \begin{bmatrix} 1 \\ \frac{\mathbf{v}^\mathsf{T}\mathbf{1}_n}{n} \end{bmatrix} + o(1)
$$

where we use Theorem 1 for the second line and recall that the asymptotic location $\lambda$ of $\hat{\lambda}$ is away from the support of limiting spectral measure $\omega$ so that $-\frac{1}{2\pi\imath} \oint_\Gamma m(z)\, dz = 0$ in the third line.

Interestingly, note at this point that taking $\mathbf{v}^\mathsf{T} \mathbf{1}_n = 0$ or $a_2 = 0$ (so that $\Omega(z) = 0$) leads to the same simplification

$$
\frac{1}{n} |\mathbf{v}^\mathsf{T} \hat{\mathbf{v}}|^2 = \lim_{z \to \lambda} (z - \lambda)\Theta(z)m^2(z) + o(1) = \lim_{z \to \lambda} (z - \lambda)\frac{a_1\rho m^2(z)}{c + a_1 m(z)(1 + \rho)} + o(1) \tag{34}
$$

$$
= \frac{a_1\rho}{1+\rho} \frac{m^2(\lambda)}{m'(\lambda)} + o(1) = \frac{a_1\rho}{1+\rho} \left( 1 - \frac{a_1^2 c m^2(\lambda)}{(c + a_1 m(\lambda))^2} - \frac{\nu - a_1^2}{c} m^2(\lambda) \right) + o(1) \tag{35}
$$

with l'Hospital's rule and the fact that $m'(z) = \left( \frac{1}{m^2(z)} - \frac{a_1^2 c}{(c + a_1 m(z))^2} - \frac{\nu - a_1^2}{c} \right)^{-1}$ by differentiating (9). The particularly means that, in the absence of the (noisy) non-informative spikes due to $a_2 \neq 0$ *or* in the case of balanced class $\mathbf{v}^\mathsf{T} \mathbf{1}_n = 0$ (in fact "almost" balanced class $\mathbf{v}^\mathsf{T} \mathbf{1}_n = o(n)$), we obtain the *same* asymptotic alignment (with respect to $\mathbf{v}$) for the *informative* eigenvector. However, there may appear spurious non-informative and isolated eigenvectors in the latter case.

In the setting of Corollary 2 with $a_1 > 0$ and $a_2 = 0$, with the substitution $m(\lambda) = m_\rho = -\frac{c}{a_1(\rho+1)}$ into (35) and then the change-of-variable $m = -\frac{c}{a_1}\frac{1}{1+x}$, we obtain the expression of $F(x)$ in Corollary 2. The phase transition condition can be similarly obtained, as discussed in Section A.2, by checking the sign of the derivative of the functional inverse $x'(m)$ as in Corollary 3. This concludes the proof of Corollary 2. $\qquad \square$

**Discussions.** Note that, while with either $a_2 = 0$ or $\mathbf{v}^\mathsf{T}\mathbf{1}_n = 0$ we obtain the same expression for the projection $|\mathbf{v}^\mathsf{T}\hat{\mathbf{v}}|^2$, the possible spike of interest $\hat{\lambda}$ (and its asymptotic location $\lambda$) in these two scenarios can be rather different. More precisely,

1. with $a_2 = 0$, there is a single possible spike $\hat{\lambda}$ with $m(\lambda) = m_\rho = -\frac{c}{a_1(\rho+1)}$;

2. with $\mathbf{v}^\mathsf{T}\mathbf{1}_n = 0$, there can be up to three spikes that correspond to $m_\rho = -\frac{c}{a_1(\rho+1)}$ and
   $m_\pm = \pm\frac{c}{a_2}\sqrt{\frac{2}{\kappa-1}}$.

This observation leads to the following remark.

**Remark 4** (Noisy top eigenvector with $a_2 \neq 0$). *For $\mathbf{v}^\mathsf{T}\mathbf{1}_n = 0$ and $a_2 \neq 0$, one may have $m_- = -\frac{c}{a_2}\sqrt{\frac{2}{\kappa-1}} > -\frac{c}{a_1(\rho+1)} = m_\rho$ for instance with large $a_2$ and small $a_1$. Since $m(x)$ is an increasing function, the top eigenvalue-eigenvector pair of $\mathbf{K}$ can be non-informative, independent of the SNR $\rho$, and totally useless for clustering purposes. An example is provided in Figure 2 where one observes that (**i**) the largest spike (on the right-hand side) corresponds to a noisy eigenvector while the second largest spike has its eigenvector positively aligned with the label vector $\mathbf{v}$; and (**ii**) the theoretical prediction of the eigen-alignment $\alpha$ in Corollary 2 still holds here due to $\mathbf{v}^\mathsf{T}\mathbf{1}_n = 0$. This extends our Corollary 1 to the signal-plus-noise scenario and confirms the advantage and necessity of taking $a_2 = 0$.*

As a side remark, in contrast to Remark 3 and Figure 13, where we observe that the non-informative spike can be lying between the two disjoint regions of the limiting measure $\omega$, in the case of $a_1 > 0$, the informative spike $m_\rho = -\frac{c}{a_1(\rho+1)}$ can *only* appear on the *right-hand side* of the support of $\omega$, since $-\frac{c}{a_1} < -\frac{c}{a_1(\rho+1)} < 0$ for $\rho = \lim_p \|\boldsymbol{\mu}\|^2 \geq 0$. See Figure 8-(bottom) for an illustration.

### A.4 PROOF OF PROPOSITION 1

Note that, for $\hat{\mathbf{v}}$ the top isolated eigenvector of $\mathbf{K}$, with Corollary 2 we can write

$$\hat{\mathbf{v}} = \sqrt{\alpha}\mathbf{v}/\sqrt{n} + \sigma\mathbf{w} \tag{36}$$

for some $\sigma \in \mathbb{R}$, $\mathbf{w} \in \mathbb{R}^n$ a zero-mean random vector, orthogonal to $\mathbf{v}$, and of unit norm. To evaluate the asymptotic clustering performance in the setting of Proposition 1 (i.e., with the estimate $\hat{\mathcal{C}}_i = \text{sign}([\hat{\mathbf{v}}]_i)$ for $\hat{\mathbf{v}}^\mathsf{T}\mathbf{v} \geq 0$), we need to assess the probability $\Pr(\text{sign}([\hat{\mathbf{v}}]_i) < 0)$ for $\mathbf{x}_i \in \mathcal{C}_1$ and $\Pr(\text{sign}([\hat{\mathbf{v}}]_i) > 0)$ for $\mathbf{x}_i \in \mathcal{C}_2$ (recall that the class-label $[\mathbf{v}]_i = -1$ for $\mathbf{x}_i \in \mathcal{C}_1$ and $[\mathbf{v}]_i = +1$ for $\mathbf{x}_i \in \mathcal{C}_2$), and it thus remains to derive $\sigma$. Note that

$$1 = \hat{\mathbf{v}}^\mathsf{T}\hat{\mathbf{v}} = \alpha + 2\sigma\sqrt{\alpha}\mathbf{w}^\mathsf{T}\mathbf{v}/\sqrt{n} + \sigma^2 = \alpha + \sigma^2 + o(1) \tag{37}$$

where we recall $\|\mathbf{v}\| = \sqrt{n}$, which, together with an argument on the normal fluctuations of $\hat{\mathbf{v}}$ (Kadavankandy & Couillet, 2019), concludes the proof. $\square$

## B ADDITIONAL EMPIRICAL RESULTS ON REAL-WORLD DATASETS

In Figure 14, we compare the clustering performance, the level of sparsity, and the computational time of the top eigenvector, of the sparse function $f_1$ and quantized $f_2$ with $M = 2$ (so 2 bits per nonzero entry), on the MNIST dataset. We see that, different from the binary $f_3$ with which small entries of $\mathbf{K}$ are set to zero, the quantized function $f_2$, by letting the small entries of $\mathbf{K}$ to take certain *nonzero* values, yields surprisingly good performance on the MNIST dataset. This performance gain comes, however, at the price of somewhat heavy computational burden that is approximately the same as the original dense matrix $\mathbf{X}^\mathsf{T}\mathbf{X}$, since we lose the sparsity with $f_2$, see Figure 1-(left). This may be interpreted as a trade-off between storage size and computational time.

Also, from the left and middle displays of Figure 7 and Figure 14, we see that for MNIST data, while the classification error rates on the pair of digits $(0, 1)$ can be as low as $1\%$, the performances on the pair $(5, 6)$ are far from satisfactory, with the linear $f(t) = t$ and the proposed $f_1$, $f_2$ or $f_3$. This is the limitation of the proposed statistical model in (1), which only takes into account the *first-order discriminative statistics*. Indeed, it has been shown in (Liao & Couillet, 2019) that, taking $a_2 = 0$ (as

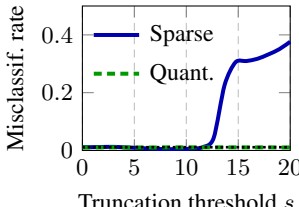 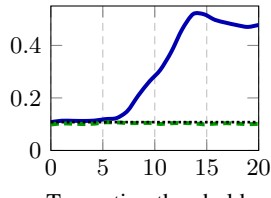 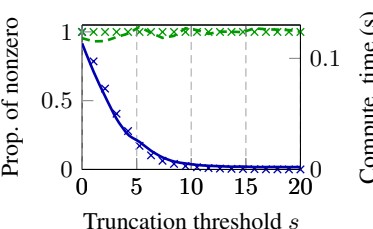

Figure 14: Clustering performance (**left**), proportion of nonzero entries, and computational time of the top eigenvector (**right**, in markers) of sparse $f_1$ and quantized $f_2$ with $M = 2$, as a function of the truncation threshold $s$ on the MNIST dataset: digits $(0, 1)$ (**left**) and $(5, 6)$ (**middle** and **right**) with $n = 2\,048$ and performance of the linear function in **black**. Results averaged over 100 runs.

in the case of the proposed $f_1$, $f_2$ and $f_3$) asymptotically discards the *second-order discriminative statistics* in the covariance structure, and may thus result in suboptimal performance in the case of non-identity covariance. It would be of future interest to extend the current analysis to the "generic" Gaussian mixture classification: $\mathcal{N}(\boldsymbol{\mu}_1, \mathbf{C}_1)$ versus $\mathcal{N}(\boldsymbol{\mu}_2, \mathbf{C}_2)$ by considering the impact of (**i**) asymmetric means $\boldsymbol{\mu}_1$ and $\boldsymbol{\mu}_2 \neq -\boldsymbol{\mu}_1$ and (**ii**) statistical information in the covariance structure $\mathbf{C}_1$ versus $\mathbf{C}_2$ and (**iii**) possibly a multi-class mixture model with number of classes $K \geq 3$.

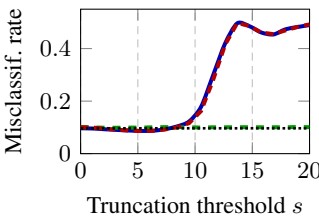 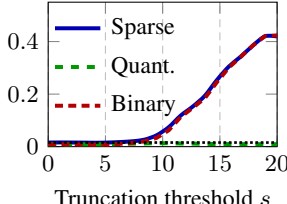 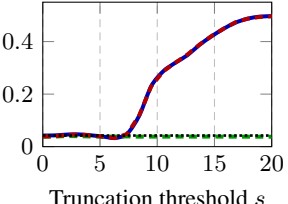

Figure 15: Clustering performance of sparse $f_1$, quantized $f_2$ (with $M = 2$) and binary $f_3$ as a function of the truncation threshold $s$ on: (**left**) Kuzushiji-MNIST class 3 versus 4, (**middle**) Fashion-MNIST class 0 versus 9, and (**right**) Kannada-MNIST class 4 versus 8, for $n = 2\,048$ and performance of the linear function in **black**. Results averaged over 100 runs.

Figure 15 compares the clustering performances of the proposed $f_1$, $f_2$, and $f_3$ on other MNIST-like datasets including the Fashion-MNIST (Xiao et al., 2017), Kuzushiji-MNIST (Clanuwat et al., 2018), and Kannada-MNIST (Prabhu, 2019) datasets. Then, Figure 16 compares the performances on the representations of the ImageNet dataset (Deng et al., 2009) from the popular GoogLeNet (Szegedy et al., 2015) of feature dimension $p = 2\,048$. On various real-world data or features, we made similar observations as in the case of MNIST data in Figure 7 and Figure 14: the performances of sparse $f_1$ and binary $f_3$ are very similar and generally degrade as the threshold $s$ becomes large, while the quantized $f_2$ yields consistently good performances that are extremely close to that of the linear function. This is in line with the (theoretically sustained) observation in (Seddik et al., 2020) that the "deep" representations of real-world datasets behave, in the large $n, p$ regime, very similar to simple Gaussian mixtures, thereby conveying a strong practical motivation for the present analysis.

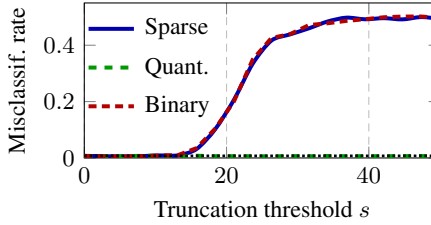 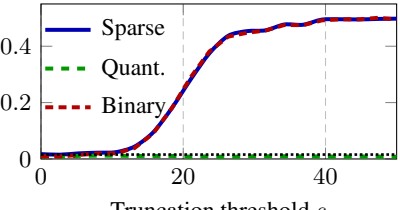

Figure 16: Clustering performance of sparse $f_1$, quantized $f_2$ (with $M = 2$) and binary $f_3$ as a function of the truncation threshold $s$ on *GoogLeNet* features of the ImageNet datasets: (**left**) class "pizza" versus "daisy" and (**right**) class "hamburger" versus "coffee", for $n = 1\,024$ and performance of the linear function in **black**. Results averaged over 10 runs.

