# OpenReview forum: "Sparse Quantized Spectral Clustering"
_ICLR.cc/2021/Conference — ICLR 2021 Spotlight_

### Official Review · AnonReviewer3 · 2020-10-27
**Good work, but presentation too technical**

**Rating:** 7
**Confidence:** 2

**Review:**

The paper rigorously studies the effect of sparsification and quantisation on the eigen-spectrum of kernel matrices, and subsequently on downstream tasks, such as spectral clustering. It is shown both theoretically and numerically that significant reduction in computation can be introducing entry wise non-linear operations (sparsification etc) on the kernel matrix with significant reduction in performance.

The paper is of high technical quality, but the presentation is too technical, making the paper difficult to follow (particularly Section 3 and parts of Section 4). Moreover, there is hardly any discussion on the limitations of the analysis. The study makes few restrictions: the data is assumed to be sampled from two high-dimensional sub-gaussian mixtures and kernel function is assumed to a function of the dot product of the data. The latter assumption is somewhat restrictive since dot-product type kernels are not the popular choice, at least, not for spectral clustering.
The authors may be interested in a work on high-dimensional clustering with dot-product kernels, which is based on concentration inequalities and hence can provide finite sample analysis [Vankadara and Ghoshdastidar, AISTATS 2020]. The work, however, does not focus on sparsification etc.

---

> ### Author Response · Authors · 2020-11-14
> **Response to Review 3**
>
> We thank the reviewer for his/her time reviewing our work and for the pertinent and constructive comments on our paper.
>
> In line with our comments to Reviewer 7 and 5, we agree that the article would heavily gain in readability if our core messages are simplified, beyond the mere exposition of our (possibly seen as too technical) results. In the revised version of the article, we will heavily reorganize Section 3 and 4 by adding more explanations and discussions, so as to better convey our main message that spectral clustering, and more generally kernel-based machine learning methods as a whole, are stable under possibly heavy sparsification and/or quantization (this conclusion following on theoretical standpoint from Corollary 2 and Proposition 1, and on an application angle from the experiments in Figure 7 and Appendix B). The revised version will also be the opportunity for us to discuss the limitations of our proposed analysis (some of these discussions are already present in Appendix B due to space limitation but will be moved back to the core article), and to cite and discuss the sharper (non-asymptotic) results in [Vankadara and Ghoshdastidar, AISTATS 2020].

---

### Official Review · AnonReviewer5 · 2020-11-08
**analysis of an important case of a natural model**

**Rating:** 7
**Confidence:** 3

**Review:**

This paper studies algorithms for dropping entries from a dense Gram matrix constructed from a lower rank, p-by-n X matrix, with n much larger than p. It analyzes the preservation of eigenstructures, as well as the overall performance of clustering algorithms when the columns of X are random vectors, and the sparsity of the matrix is reduced by one of three methods: keeping only large magnitude entries, 0/1 based on whether magnitude exceeds a threshold, or a more continuous version between the 0 and 1 (in the small magnitude range). Some experimental comparisons with other sparsification methods such as uniform sampling are made.

Strengths:
+ The problem studied is an important one for analyzing large datasets.
+ The study of different  'filter' functions on the sampled entries is quite novel to the matrix sampling literature
+ The criteria used for measuring quality of output is holistic, and in my opinion significant step forward from simply measuring errors in the matrix sense.

Weaknesses:
- The randomness assumption of all columns of X being independent unit vectors is somewhat generous.
- Computing eigenvectors of a p-by-n X matrix can be done by solving p-by-p linear systems involving the matrix XX^T instead, and for many reasonable notions of errors, this matrix can be sketched. However, I do feel the analysis done is more broadly applicable.

Overall, I feel this paper takes an interesting step toward problem specific, and algorithmic specific sparsification. The set up is natural, and the mathematical depth is significant. Overall, the only main drawback I can think of is I had trouble extracting a higher level message from the formal statements, but I believe the results are more than sufficient for a paper.

---

> ### Author Response · Authors · 2020-11-14
> **Response to Review 5**
>
> We would first like to thank the reviewer for the time taken to review our paper and for his/her positive support as well as the constructive comments.
>
> In this article, while $n$ is clearly (much) larger than $p$, we position ourselves here under the high-dimensional regime where the data dimension $p$ is also large: this has been shown in repeated studies recently (see e.g., [V18, PW17, LLC18, BHMM19]) to be a more realistic assumption to tackle machine learning models of not-too-small data dimensions (typically of size $p>100$).
>
> In the revised version, we will better emphasize our high-level message which states that, when high-dimensional data (with some natural statistical properties as per (1)) are involved, the eigen-structure of $X^T X$ can be preserved even under very aggressive sparsification and/or quantization (this being confirmed theoretically and compellingly supported by our experiments in Figure 7 and Appendix B on real data of various nature).
>
> [V18] Vershynin, R., 2018. High-dimensional probability: An introduction with applications in data science (Vol. 47). Cambridge university press.
> [PW17] Pennington, J. and Worah, P., 2017. Nonlinear random matrix theory for deep learning. In Advances in Neural Information Processing Systems (pp. 2637-2646).
> [LLC18] Louart, C., Liao, Z. and Couillet, R., 2018. A random matrix approach to neural networks. The Annals of Applied Probability, 28(2), pp.1190-1248.
> [BHMM19] Belkin, M., Hsu, D., Ma, S. and Mandal, S., 2019. Reconciling modern machine-learning practice and the classical bias–variance trade-off. Proceedings of the National Academy of Sciences, 116(32), pp.15849-15854.

---

### Official Review · AnonReviewer7 · 2020-11-10
**Paper seems interesting and tackles a very important topic, a bit hard to unpack.**

**Rating:** 6
**Confidence:** 3

**Review:**

This is a nice paper that shows that one can perturb a kernel matrix (or pass it through a non-linear transformation) without necessarily modifying the underlying eigenspectrum significantly, and as such, without hurting the performance of spectral clustering applied on the matrix. The most important application I can immediately see is sparsifying the kernel matrix so it can be computationally used efficiently. Or similarly, apply quantization and binarization, as the authors mention.

First, a high level point: I wonder what the implication is of a non-linear transformation applied on X*X' instead of the design matrix itself X -- which is what seems more useful to me. You would want to apply the non-linear transformation on the actual vectors, not on the dot products between all of their pairs, isn't it? Though, I do see the value in "zeroing out", for example, or sparsifying the kernel matrix X*X', as working with kernel matrices can be problematic because of their size and density. I suppose, also, maybe a transformation on X can be translated into a transformation on X*X'? Not always.

I also wonder if the model presented here is not too simple to make the claim that spectral clustering is stable under perturbation to the kernel matrix. My understanding is that the author's model is rather toyish, consisting of only two clusters -- is that correct? How do your results generalize to more than two clusters? Is it a trivial step to make, or does it require significantly more thought?

Regarding Assumption 1, it is quite difficult to unpack it. Can you at least confidently state that Eq.3-5 satisfy this assumption? Can you give a few examples of other f's that satisfy that, or at least give an intuition what this assumption means? What properties of f is it related to? How is p coming into play?

Theorem 1 again needs some unpacking and intuition. Why is it important? The text that comes after the theorem makes it even more confusing -- what is the implication on the K matrix for that? For example, how would it be reduced for f which is just a sparsification, I would be curious? Is there a way to instantiate all the quantities in Theorem 1 as a corollary to get a better picture of how sparsification affects K and how it differs from previous results in the literature? Figure 1 helps do that, but it is far from being complete.

Proposition 1 is a bit easier to unpack (though "with the convention... etc" maybe add a footnote that there is some degree of freedom there, otherwise it is unclear). I am slightly concerned that under all these complex expressions there might be results that are not very strong, but it is hard to track, because tracing each quantity back to a concrete value would take a lot of time. Again, I strongly urge the authors to instantiate some of their results to simple cases, where many of the variables and traces of variables disappear, so that the strength of the results can be thought about.

I would expect for example a corollary in a simple form that roughly states: "Say x is drawn from the clustering model as above in Eq. 1. Say we use f as in Eq. 3 (and hence Assumption 1 is instantiated with the orthogonal polynomial framework). Then: (a) the eigenvector/eigenvalue of K stays such and such within the one of X; (b) spectral clustering will have this much or such increase in the error rate."

If you could do it for f's in Eq. 4-5 that's even better.

Minor comments:

1. "One approach to overcoming this limitation is simple subsampling: dividing X into subsamples of size εn, for some ε ∈ (0, 1), on which one performs parallel computation, and then recombining."

What type of computation?

2. In Eq. 1 -- are you saying you have a mixture model with two components C1 and C2? Might be a good idea to state that explicitly. Is there a specific prior probability to belong to C_1 or C_2?

3. Explain what are "isolated eigenvalues" or cite.


Full disclosure: given the time constraints and me being an emergency reviewer, I didn't look carefully at the supplementary materials.

---

> ### Author Response · Authors · 2020-11-14
> **Response to Review 7**
>
> We would first like to thank the reviewer for his/her positive support and for the thorough and helpful remarks.
>
> **Non-linear transformation on X**
> We agree with the reviewer that performing the non-linear transformation directly on $X$ is another natural setting, since this reduces the complexity of, in the first place, computing the matrix product $X^T X$. The eigenspectrum study of $f(X)^T f(X)$, and specifically of $f(W X)^T f(W X)$ for some random projection matrix $W$ has been performed on the case where $X$ has i.i.d. (noise-like) entries [PW17]; we believe that our analysis can be extended to those models, on more structural and realistic data.
>
> **Two-cluster model**
> We present our results on a (symmetric) two-cluster model here mainly for simplicity of exposition and interpretable ease. Our results straightforwardly generalize to a multi-cluster setting under a more general mixture model as in (Liao & Couillet, 2019). This, however, may unavoidably lead to even more complex results than, e.g., the current Corollary 1 and Proposition 1. In the current version of the article, we provided experiments on MNIST data in Figure 7, and on more MINIST-like data as well as on CNN features of ImageNet data in Appendix B (those having been moved there due to space limitation). These empirical results, together with our theoretical results on a natural, yet already non-trivial, model, convey our main message that spectral clustering and more generally kernel-based methods are stable under non-linear transformation such as sparsification and quantization. We will clarify the possible extensions, and place more emphasis on the experimental results in the revised version of this article.
>
> **Assumption 1**
> Assumption 1 is merely a technical assumption here, posed to ensure that our technical result in Theorem 1 holds for a larger family of non-linear function $f$ that may be of broader interest (beyond those in Eq.3-5). Note that, in the current version, we state the $f_1, f_2, f_3$ under consideration *all* satisfy Assumption 1 (just below Figure 1). In effect, the vast majority of commonly used non-linear functions satisfy Assumption 1 (counter-example exists though, e.g., $\exp(P(t))$ fails for any polynomial $P(t)$ of degree greater than $2$), and it is stated here mainly for technical completeness.
> Still, we agree with the reviewer that the assumption, as it stands, looks heavily technical and hides the, after all, 'mild' nature of the request on $f$; in our revised version, we will clarify this discussion and introduce the aforementioned counter-example for concreteness.
>
> **Theorem 1**
> The importance of Theorem 1 is briefly discussed after (6): as a reminder, it can be shown (in Corollary 2) and confirmed in Figure 2 that only isolated eigenvectors contain the cluster information. These isolated eigenpairs appear to be a key object in spectral clustering analysis [Baik et al., 2005, VL07, JY16], and refer to the two rightmost eigenvalues visually *away* from the main eigenvalue bulk in Figure 2. We will add more explanations and citations to clarify this in a revised version.
> As a consequence, to qualitatively evaluate the performance of spectral clustering, one needs to discover the limiting spectrum of the main bulk (red dashed line in Figure 2) as well as the behavior of isolated eigenvectors (Corollary 2): these all follow from the key Theorem 1, as shortly discussed below (6). The major implication of Theorem 1 is that the aforementioned spectral behavior of $K$ depends on the non-linear $f$ only via the *three* parameters $a_1, a_2, \nu$ defined in (8). In particular, for $f_1, f_2, f_3$, we have $a_2 = 0$ and it remains to discuss the ratio $a_1^2/\nu$ as has been done in Section 4.
> We see that these unconventional notions are prone to mislead the readers and will add more detailed explanations of our technical results such as Theorem 1 in the revised version of the article.
>
> **Proposition 1**
> As mentioned above for Theorem 1, the eigenspectrum of $K$ depends on $f_1, f_2, f_3$ via the corresponding $a_1^2/\nu$, which, via $F(x)$ defined in (10), determines the eigen-alignment $\alpha$ in (11) and therefore the spectral clustering performance via Proposition 1. This connection is discussed after Remark 1 and illustrated in Figures 3 and 4. Again, we see that these notions, classical in random matrix analysis, are possibly less familiar to machine learning experts not used to these technical terms: we will better explain the necessary connections in the revised version of the article.
>
> [PW17] Pennington, J. and Worah, P., 2017. Nonlinear random matrix theory for deep learning. In Advances in Neural Information Processing Systems (pp. 2637-2646).
> [VL07] Von Luxburg, U., 2007. A tutorial on spectral clustering. Statistics and computing, 17(4), pp.395-416.
> [JY16] Joseph, A. and Yu, B., 2016. Impact of regularization on spectral clustering. The Annals of Statistics, 44(4), pp.1765-1791.

---

### Official Review · AnonReviewer6 · 2020-11-10
**The paper derives precise asymptotic results for pre-processing steps used in machine learning problems to boost computational efficiency, resulting in a complete asymptotic characterization of tradeoffs.**

**Rating:** 7
**Confidence:** 4

**Review:**

This work considers the effect of sparsification, quantization and non-linear transormations on the spectrum of a random matrix with respect to performance in downstream applications like clustering. Eigen decomposition of large matrices is computationally very expensive and therefore methods like sparsification is used in order to reduce the computational complexity, while adding some amount of error. Therefore, theoretical bounds which quantize the amount of deviation because of such processing steps is essential to understand how much/little degraded the performance of machine learning applications are because of this step. This work considers the important case of spectral clustering with a 2 class mixture of subexponential models using
the Grammian matrix X^{T}X under entry wise non-linear transformations (specifically thresholding, binarization and quantization).

The theoretical analysis derives asymptotic eigenvalue distribution f(X^{T}X/sqrt{p})/sqrt{p} for some entry wise non linear function f. The analysis heavily relies on entry wise central limit theorems for each entry and expansion of
f using Gaussian orthogonal polynomials. It is shown that the limiting distribution depends only on the low order coefficients in the Gaussian polynomial expansion. Asymptotic accuracy rates for spectral clustering follows via Cauchy integral formula applied to the resolvent matrix, which is a standard technique in random matrix theory.

Proposition 1 is the most relevant contribution of the paper to machine learning - a parameter gamma is recognized based on the polynomial expansion of f which is a threshold for signal to noise ratio rho. if rho < gamma, asymptotically,
spectral clustering doesn't work and when rho > gamma, precise asymptotic accuracy is derived for spectral clustering, depending only on rho and f. The analysis is largely independent on the specific mixture distribution considered because of the
use of central limit theorems.


The results are very nice and gives a firm theoretical understanding of a well studied practical problem in machine learning. The results are precise and look complete. I would like to see further research with non-asymptotic rates for misclassification.
It could be possible to apply quantitative Berry-Esseen type local limit theorems to derive such results. This clearly deserves to be published in a venue such as ICLR
Some minor comments:

1.  it is called data dependent but notation f(M/sqrt{p})sqrt{p} doesn't suggest data-dependent quantization. How is this data dependent? the setting of parameter s in the definition of f?
2. line 10, page 2 - corroborate instead of collabora

---

> ### Author Response · Authors · 2020-11-14
> **Response to Review 6**
>
> We would first like to thank the reviewer for his/her positive support and for the time reviewing our work.
>
> We refer to our approach as “data-dependent” since, with some prior knowledge on the data (for instance, the signal-to-noise ratio $\rho = \lim \| \mu \|^2$ defined in Corollary 2), the nonlinear function $f$ can be designed (e.g., by choosing the threshold $s$ and the number of information bits $M$, see Figure 1) to achieve a better performance-complexity trade-off for spectral clustering (and presumably for many other algorithms). As a concrete example, in the right plot of Figure 5, one is able to “compress” the model much more aggressively at a high SNR, with virtually no performance loss. We note that this term may cause unnecessary confusion. In an updated version of this article, we will revise this term, and add more discussions on the limitations of our proposed analysis and on future extensions to (sharper) non-asymptotic results.

---

### Decision · Program_Chairs · 2021-01-07
**Final Decision**

**Decision:**

Accept (Spotlight)

**Comment:**

The paper presents a nice analysis of the spectrum of a matrix that is obtained by applying non-linear functions to a random matrix. The paper is mostly well-written, the result is novel and interesting, and has clear implications for ML problems like spectral clustering.
So I would enthusiastically recommend the paper for acceptance at ICLR.
It would be important for authors to take into account reviewer comments. In particular, instantiating the theorems for simple ML-centric examples would be very useful.